# BETTER IMITATION LEARNING IN DISCOUNTED LINEAR MDP

## ABSTRACT

We present a new algorithm for imitation learning in infinite horizon linear MDPs dubbed ILARL which greatly improves the bound on the number of trajectories that the learner needs to sample from the environment. In particular, we remove exploration assumptions required in previous works and we improve the dependence on the desired accuracy $\epsilon$ from $\mathcal{O}\left(\epsilon^{-5}\right)$ to $\mathcal{O}\left(\epsilon^{-4}\right)$. Our result relies on a connection between imitation learning and online learning in MDPs with adversarial losses. For the latter setting, we present the first result for infinite horizon linear MDP which may be of independent interest. Moreover, we are able to provide a strengthen result for the finite horizon case where we achieve $\mathcal{O}\left(\epsilon^{-2}\right)$. Numerical experiments with linear function approximation shows that ILARL outperforms other commonly used algorithms.

## 1 INTRODUCTION

Imitation Learning (IL) is of extreme importance for all applications where designing a reward function is cumbersome while collecting demonstrations from an expert policy $\pi_E$ is easy. Examples are autonomous driving Knox et al. (2021), robotics Osa et al. (2018), and economics/finance Charpentier et al. (2020). The goal is to learn a policy which competes with the expert policy under the true unknown cost function of the Markov Decision Process (MDP) Puterman (1994).

Imitation learning relies on two data resources: expert demonstrations collected acting with $\pi_E$ and data that can be collected interacting in the MDP with policies chosen by the learning algorithm. The first approach known as behavioural cloning (BC) solves the problem applying supervised learning. That is, it requires no interaction in the MDP but it requires knowledge of a class $\Pi$ such that $\pi_E \in \Pi$ and $\widetilde{\mathcal{O}}\left(\frac{\log|\Pi|}{(1-\gamma)^4 \epsilon_E^2}\right)$ expert demonstrations to ensure with high probability that the output policy is at most $\epsilon_E$-suboptimal.

The quartic dependence on the effective horizon term $((1-\gamma)^{-1})$ is problematic for long horizon problems. Moreover, the dependence on $\Pi$ requires to make prior assumption on the expert policy structure to provide bounds which do not scale with the number of states in the function approximation setting. Thankfully, the dependence on the effective horizon can be improved resorting to MDP interaction. There exists an interesting line of works achieving this goal considering an interacting setting where the learner has the possibility to query the expert policy at any state visited during the MDP interaction Ross & Bagnell (2010); Ross et al. (2011) or that require a generative model to implement efficiently the moment matching procedure Swamy et al. (2022). Another recent work requires a generative model to sample the initial state of the trajectory from the expert occupancy measure Swamy et al. (2023). In this work, we considered a different scenario which is adopted in most of applied imitation learning Ho et al. (2016); Ho & Ermon (2016); Fu et al. (2018); Reddy et al. (2019); Dadashi et al. (2021); Watson et al. (2023); Garg et al. (2021). In this case, the expert policy can not be queried but only a dataset of expert demonstrations collected beforehand is available.

The setting has received scarse theoretical attention so far. The only results we are aware of are: Shani et al. (2021) that focus on the tabular, finite horizon case, Liu et al. (2022) in the finite horizon linear mixture MDP setting and Viano et al. (2022) in the infinite horizon Linear MDP setting. In all these works bound the number of required expert demonstrations scale as $(1-\gamma)^{-2}$ which improves considerably over the quartic depedence attained by BC. However, Viano et al. (2022) made the following assumption on the features that greatly simplifies the exploration in the MDP.

Table 1: **Comparison with related algorithms** Our algorithms provide guarantees for the number of expert trajectories independent on $\mathcal{S}$ and $\Pi$ without assumptions on the expert policy. For what concerns, the MDP trajectories we provide the best known results in finite and infinite horizon linear MDPs. By **Linear Expert**, me mean that the expert policy is $\pi(s) = \max_{a \in \mathcal{A}} \phi(s,a)^{\mathsf{T}} \theta$ for some unknown vector $\theta$.

| Algorithm | Setting | Expert Traj. | MDP Traj. |
|---|---|---|---|
| Behavioural Cloning | Function Approximation, Offline Agarwal et al. (2019) | $\mathcal{O}\left(\frac{H^4 \log|\Pi|}{\epsilon^2}\right)$ | - |
| | Tabular, Offline Rajaraman et al. (2020) | $\widetilde{\mathcal{O}}\left(\frac{H^2|\mathcal{S}|}{\epsilon}\right)$ | - |
| | Linear Expert, Offline Rajaraman et al. (2021) | $\widetilde{\mathcal{O}}\left(\frac{H^2 d}{\epsilon}\right)$ | - |
| Mimic-MD Rajaraman et al. (2020) | Tabular, Known Transitions, Deterministic Expert | $\mathcal{O}\left(\frac{H^{3/2}|\mathcal{S}|}{\epsilon}\right)$ | - |
| OAL Shani et al. (2021) | Tabular | $\mathcal{O}\left(\frac{H^2|\mathcal{S}|}{\epsilon^2}\right)$ | $\mathcal{O}\left(\frac{H^4|\mathcal{S}|^2|\mathcal{A}|}{\epsilon^2}\right)$ |
| MB-TAIL Xu et al. (2023) | Tabular, Deterministic Expert | $\mathcal{O}\left(\frac{H^{3/2}|\mathcal{S}|}{\epsilon}\right)$ | $\mathcal{O}\left(\frac{H^3|\mathcal{S}|^2|\mathcal{A}|}{\epsilon^2}\right)$ |
| OGAIL Liu et al. (2022) | Linear Mixture MDP | $\mathcal{O}\left(\frac{H^3 d^2}{\epsilon^2}\right)$ | $\mathcal{O}\left(\frac{H^4 d^3}{\epsilon^2}\right)$ |
| PPIL Viano et al. (2022) | Linear MDP, Persistent Excitation | $\mathcal{O}\left(\frac{d}{(1-\gamma)^2 \epsilon^2}\right)$ | $\mathcal{O}\left(\frac{d^2}{\beta^6(1-\gamma)^9 \epsilon^5}\right)$ |
| **ILARL** (Algorithm 3) | Linear MDP | $\mathcal{O}\left(\frac{d}{(1-\gamma)^2 \epsilon^2}\right)$ | $\mathcal{O}\left(\frac{d^3}{(1-\gamma)^8 \epsilon^4}\right)$ |
| **BRIG** (Algorithm 4) | Episodic Linear MDP | $\mathcal{O}\left(\frac{dH^2}{\epsilon^2}\right)$ | $\mathcal{O}\left(\frac{d^3 H^4}{\epsilon^2}\right)$ |

**Assumption. Persistent excitation**     *It holds that for any policy $\pi^k$ in the sequence of policies generated by the algorithm adopted by the learner* $\lambda_{\min}\left(\mathbb{E}_{s,a \sim d^{\pi^k}}\left[\phi(s,a)\phi(s,a)^{\mathsf{T}}\right]\right) \geq \beta > 0$.

Despite being commonly used in infinite horizon function approximation setting (see for example Abbasi-Yadkori et al. (2019a); Hao et al. (2021); Duan et al. (2020); Lazic et al. (2020); Abbasi-Yadkori et al. (2019b); Agarwal et al. (2020a)), the persistent excitation assumption is very restrictive as it can be easily violated by deterministic policies with tabular features.

**Our contribution**     We propose a new algorithm that improves the results of Viano et al. (2022) in two important aspects: it **bypasses the persistent excitation assumption** (i.e. $\beta = 0$ does not cause the bound to blow up) and **it improves the dependence on** $\epsilon$. In particular, the new proposed algorithm Algorithm 3 only requires $\mathcal{O}\left(\frac{d^3}{(1-\gamma)^8 \epsilon^4}\right)$ MDP interactions which greatly improves upon the bound $\widetilde{\mathcal{O}}\left(\frac{d^2}{\beta^6(1-\gamma)^9 \epsilon^5}\right)$ proven by Viano et al. (2022).

The design is different from Viano et al. (2022) and it builds on a connection between imitation learning and online learning in MDP with full information. Therefore, **we design** as a submodule of our algorithm **the first algorithm for adversarial infinite horizon linear MDPs** which achieves $\mathcal{O}(K^{3/4})$ pseudo-regret. We also consider the finite horizon version of this algorithm which obtains a regret bound $\widetilde{\mathcal{O}}\left(d^{3/4} H^{3/2} K^{3/4}\right)$ which improves by a factor $H^{1/2}$ the first result in this setting proven in Zhong & Zhang (2023). Concurrently to our work Sherman et al. (2023a) derived a further improvement with optimal dependence on $K$.

Finally, **we provide a stronger result for the finite horizon setting**. Key for this result is realizing that in the regret decomposition of Shani et al. (2021) one of the two players can in fact play the best response rather than a more conservative no regret strategy. This observations leads to Algorithm 4 which only requires $\mathcal{O}(H^4 d^3 \epsilon^{-2})$ MDP interactions.

**Related Works**     Early works in behavioural cloning (BC) Pomerleau (1991) popularized the framework showing its success in driving problem and Ross & Bagnell (2010); Ross et al. (2011) show that the problem can be analyzed via a reduction to supervised learning which provides an expert trajectories bound of order $\frac{H^4 \log|\Pi|}{\epsilon^2}$. In practice, it is difficult to choose a class $\Pi$ such that simultaneously contains the expert policy and is small enough to make the bound meaningful. Other algorithms like Dagger Ross et al. (2011) and Logger Li & Zhang (2022) need to query the expert interactively. In this case, the expert trajectories improve to $\frac{H^2 \max_{s,a}(A^\star(s,a))^2 \log|\Pi|}{\epsilon^2}$ where $A^\star$ is the optimal advantage. Recent works Rajaraman et al. (2020) showed that in the worst case Dagger does not improve over BC but also that both can use only $\widetilde{\mathcal{O}}\left(\frac{H^2|\mathcal{S}|}{\epsilon}\right)$ in the tabular case. Moreover,

when transitions and initial distribution are known and the expert is deterministic, the result can be improved to $\mathcal{O}\left(\frac{H^{3/2}|\mathcal{S}|}{\epsilon}\right)$ using Mimic-MD Rajaraman et al. (2020). Later, Xu et al. (2023) introduced MB-TAIL that having trajectory access to the MDP attains the same bound. This shows that the traditional bound obtained matching occupancy measure Syed & Schapire (2007) adopted in Shani et al. (2021) is suboptimal in the tabular setting. For the linear function approximation, the works in Swamy et al. (2022); Rajaraman et al. (2021) introduced algorithms that uses $\mathcal{O}\left(\frac{H^{3/2}d}{\epsilon}\right)$ expert trajectories with knowledge of the transitions but those require strong assumptions such as linear expert (Rajaraman et al., 2021, Definition 4), particular choice of features, linear reward and uniform expert occupancy measure. Rajaraman et al. (2021) also proves an improved result for BC but under the linear expert assumption which implies that the expert is deterministic. While one can notice that there exists an optimal policy in a Linear MDP which is a linear expert, in our work we do not impose assumption on the expert policy and we require $\mathcal{O}\left(\frac{H^2d}{\epsilon^2}\right)$ demonstrations. Under the same setting, the best known bound for BC is $\frac{H^2\log|\Pi|}{d}$ times larger which makes our algorithm preferrable whenever $|\Pi| \geq \exp(dH^{-2})$. We report a comparison with existing IL theory work in Table 1. As it can be noticed there is only one previous result in the infinite horizon setting Viano et al. (2022). We believe that the study of infinite horizon is important because it is the most common setting in practice Ho et al. (2016); Ho & Ermon (2016); Fu et al. (2018); Reddy et al. (2019); Dadashi et al. (2021); Watson et al. (2023); Garg et al. (2021). The practical advantage is that in the infinite horizon setting the optimal policy can be sought in the class of stationary policies which are much easier to store in memory than the nonstationary ones.

## 2 BACKGROUND AND NOTATION

In imitation learning Osa et al. (2018), the environment is abstracted as Markov Decision Process (MDP) Puterman (1994) which consists of a tuple $(\mathcal{S}, \mathcal{A}, P, c, \boldsymbol{\nu}_0)$ where $\mathcal{S}$ is the state space, $\mathcal{A}$ is the action space, $P : \mathcal{S} \times \mathcal{A} \to \Delta_{\mathcal{S}}$ is the transition kernel, that is, $P(s'|s, a)$ denotes the probability of landing in state $s'$ after choosing action $a$ in state $s$. Moreover, $\boldsymbol{\nu}_0$ is a distribution over states from which the initial state is sampled. Finally, $c : \mathcal{S} \times \mathcal{A} \to [0, 1]$ is the cost function. In the infinite horizon setting, we endow the MDP tuple with an additional element called the discount factor $\gamma \in [0, 1)$. Alternatively, in the finite horizon setting we append to the MDP tuple the horizon $H \in \mathbb{N}$ and we consider possibly inhomogenous transitions or costs function. That is, they depend on the stage within the episode. The agent plays action in the environment sampled from a policy $\pi : \mathcal{S} \to \Delta_{\mathcal{A}}$. The learner is allowed to adopt an algorithm to update the policy across episodes given the previously observed history. We will see that imitation learning has a strong connection with MDPs with adversarial costs. The latter setting allows the cost function to change each time the learner samples a new episode in the MDP. For clarity, we include the pseudocode for the interaction in Protocol 1 in Appendix B.

**Value functions and occupancy measures** We define the state value function at state $s \in \mathcal{S}$ for the policy $\pi$ under the cost function $c$ as $V^\pi(s; c) \triangleq \mathbb{E}\left[\sum_{h=1}^\infty \gamma^{h-1} c(s_h, a_h)|s_1 = s\right]$. In the finite horizon case, the state value function also depends on the stage index $h$, that is $V_h^\pi(s; c) \triangleq \mathbb{E}\left[\sum_{\ell=h}^H c(s_\ell, a_\ell)|s_h = s\right]$[1]. In both cases, the expectation over both the randomness of the transition dynamics and the one of the learner's policy. Another convenient quantity is the occupancy measure of a policy $\pi$ denoted as $d^\pi \in \Delta_{\mathcal{S} \times \mathcal{A}}$ and defined as follows $d^\pi(s, a) \triangleq (1 - \gamma) \sum_{h=1}^\infty \gamma^{h-1} \mathbb{P}\left[s, a \text{ is visited after } h \text{ steps acting with } \pi\right]$. We can also define the state occupancy measure as $d^\pi(s) \triangleq (1 - \gamma) \sum_{h=1}^\infty \gamma^{h-1} \mathbb{P}\left[s \text{ is visited after } h \text{ steps acting with } \pi\right]$. In the finite horizon setting, the occupancy measure depends on the stage $h$ and its defined simply as $d_h^\pi(s, a) \triangleq \mathbb{P}\left[s, a \text{ is visited after } h \text{ steps acting with } \pi\right]$. The state occupancy measure is defined analogously.

**Imitation Learning** In imitation learning, the learner is given a dataset $\mathcal{D}_E \triangleq \left\{\boldsymbol{\tau}^k\right\}_{k=1}^{\tau_E}$ containing $\tau_E$ trajectories collected in the MDP by an expert policy $\pi_{\mathrm{E}}$ according to Protocol 1. By trajectory $\boldsymbol{\tau}^k$, we mean the sequence of states and actions sampled at the $k^{\text{th}}$ iteration of Protocol 1, that is $\boldsymbol{\tau}^k = \left\{(s_h^k, a_k^h)\right\}_{h=1}^H$ for finite horizon case. For the infinite horizon case, the trajectories have

---

[1]In the finite horizon case we may use $V^\pi(s; c)$ as a shortcut for $V_1^\pi(s; c)$

random lenght sampled from the distribution $\text{Geometric}(1 - \gamma)$. Given, $\mathcal{D}_E$ the learner adopts an algorithm $\mathcal{A}$ to learn a policy $\pi^{\text{out}}$ such that is $\epsilon$-suboptimal according to the next definition.

**Definition 1.** *An algorithm $\mathcal{A}$ is said $\epsilon$-**suboptimal** if it outputs a policy $\pi$ whose value function with respect to the unknown true cost $\mathbf{c}_{\text{true}}$ satisfies $\mathbb{E}_{\mathcal{A}} \mathbb{E}_{s_1 \sim \boldsymbol{\nu}_0} \left[ V^\pi(s_1; \mathbf{c}_{\text{true}}) - V^{\pi_E}(s_1; \mathbf{c}_{\text{true}}) \right] \leq \epsilon$ where the first expectation is on the randomness of the algorithm $\mathcal{A}$.*

## 2.1 SETTING

We study imitation learning in the linear MDP setting popularized by Jin et al. (2019) and studied in imitation learning in Viano et al. (2022). When studying finite horizon problems we consider possible inhomogeneous transition dynamics and cost function. That is, we work under the following assumptions.

**Assumption 1.** *Episodic Linear MDP There exist a feature matrix $\boldsymbol{\Phi} \in \mathbb{R}^{|\mathcal{S}||\mathcal{A}| \times d}$ known to the learner, an unknown sequence of vectors $\mathbf{w}_h^k \in \mathbb{R}^d$ and an unknown matrix sequences $M_h \in \mathbb{R}^{d \times |\mathcal{S}|}$ such that the transition matrices $P_h$ factorize as $P_h = \boldsymbol{\Phi} M_h$ and the sequence of adversarial costs $c_h^k$ can be written as $c_h^k = \boldsymbol{\Phi} w_h^k$. Moreover, it holds for all $k \in [K], h \in [H]$ and for all state action pairs $s, a \in \mathcal{S} \times \mathcal{A}$ that $\|\boldsymbol{\Phi}\|_{1,\infty} \leq 1, \|M_h\|_{1,\infty} \leq 1, \|w_h^k\|_2 \leq 1$.*

**Assumption 2.** *Linear MDP There exist a feature matrix $\boldsymbol{\Phi} \in \mathbb{R}^{|\mathcal{S}||\mathcal{A}| \times d}$ known to the learner, an unknown sequence of vectors $\mathbf{w}^k \in \mathbb{R}^d$ and an unknown matrix $M \in \mathbb{R}^{d \times |\mathcal{S}|}$ such that the transition matrices $P$ factorize as $P = \boldsymbol{\Phi} M$ and the sequence of adversarial costs $c^k$ can be written as $c^k = \boldsymbol{\Phi} w^k$. Moreover, it holds for all $k \in [K]$ and for all state action pairs $s, a \in \mathcal{S} \times \mathcal{A}$ that $\|\boldsymbol{\Phi}\|_{1,\infty} \leq 1, \|M\|_{1,\infty} \leq 1, \|w^k\|_2 \leq 1$.*

In the context of imitation learning, we also need to assume that the true unknown cost is realizable.

**Assumption 3.** *Realizable cost The learner has access to a feature matrix $\boldsymbol{\Phi} \in \mathbb{R}^{|\mathcal{S}||\mathcal{A}| \times d}$ such that $\mathbf{c}_{\text{true}} = \boldsymbol{\Phi} w_{\text{true}}$.*

## 3 MAIN RESULTS AND TECHNIQUES

We provide our main results for the infinite horizon case in Theorem 1 and the stronger result for the finite horizon in Theorem 2.

**Theorem 1.** *Under Assumptions 2,3, there exists an algorithm, i.e. Algorithm 3, such that after using $\widetilde{\mathcal{O}} \left( \frac{\log|\mathcal{A}| d^3}{(1-\gamma)^8 \epsilon^4} \right)$ state action pairs from the MDP and using $\widetilde{\mathcal{O}} \left( \frac{2d \log(2d)}{(1-\gamma)^2 \epsilon_E^2} \right)$ expert demonstrations is $\epsilon + \epsilon_E$-suboptimal.*

**Theorem 2.** *Under Assumptions 1,3, there exists an algorithm, i.e. Algorithm 4, such that after sampling $\mathcal{O} \left( H^4 d^3 \log(dH/(\epsilon)) \epsilon^{-2} \right)$ trajectories and having access to a dataset of $\tau_E = \widetilde{\mathcal{O}} \left( \frac{2H^2 d \log(2d)}{\epsilon_E^2} \right)$ expert demonstrations is $\epsilon + \epsilon_E$-suboptimal.*

**Remark 1.** *The results are proven via the high probability bounds in Theorems 5 and 6 respectively and apply the high probability to expectation conversion lemma in Lemma 6.*

## 3.1 TECHNIQUE OVERVIEW

**Online-to-batch conversion** The core idea is to extract the policy achieving the sample complexity guarantees above via an online-to-batch conservation. That is the output policy is sampled uniformly from a collection of $K$ policies $\left\{ \pi^k \right\}_{k=1}^K$. The sample complexity result is proven, showing that the policies $\left\{ \pi^k \right\}_{k=1}^K$ produced by the algorithms under study have sublinear pseudo regret in high probability, that is,

$$\text{Regret}(K) \triangleq \frac{1}{1-\gamma} \sum_{k=1}^K \left\langle \mathbf{c}_{\text{true}}, d^{\pi^k} - d^{\pi_E} \right\rangle \leq \mathcal{O}(K^{3/4}) \quad \text{w.h.p.}$$

for the infinite horizon discounted setting with Algorithm 3 and

$$\text{Regret}(K) \triangleq \sum_{h=1}^H \sum_{k=1}^K \left\langle \mathbf{c}_{\text{true},h}, d_h^{\pi^k} - d_h^{\pi_E} \right\rangle \leq \mathcal{O}(\sqrt{K}) \quad \text{w.h.p.} \tag{1}$$

for the finite horizon setting with Algorithm 4. The next section presents the regret decomposition giving the crucial insights for the design of Algorithms 3 and 4.

**Regret decomposition**   To obtain both regret bounds, we decompose the pseudo regret in 3 terms. We present it for the infinite horizon case, where $(1 - \gamma)\mathrm{Regret}(K)$ can be upper bounded by

$$
\underbrace{\sum_{k=1}^{K}\left\langle \mathbf{c}^k, d^{\pi^k} - d^{\pi_\mathrm{E}}\right\rangle}_{\mathrm{Regret}_\pi(K; d^{\pi_\mathrm{E}})} + \underbrace{\sum_{k=1}^{K}\left\langle w_{\mathrm{true}} - w^k, \mathbf{\Phi}^\intercal d^{\pi^k} - \widehat{\mathbf{\Phi}^\intercal d^{\pi_\mathrm{E}}}\right\rangle}_{\mathrm{Regret}_w(K; w_{\mathrm{true}})} + 2\left\| \mathbf{\Phi} d^{\pi_\mathrm{E}} - \widehat{\mathbf{\Phi} d^{\pi_\mathrm{E}}}\right\|_\infty K \tag{2}
$$

This decomposition is inspired from Shani et al. (2021) but it applies also to the infinite horizon setting and exploits the linear structure using Assumptions 2,3 to write $c^k = \mathbf{\Phi} w^k$ and $\mathbf{c}_{\mathrm{true}} = \mathbf{\Phi} w_{\mathrm{true}}$.

$\mathrm{Regret}_w(K; w_{\mathrm{true}})$ is the pseudo regret of a player updating a sequence of cost functions and having $\mathbf{c}_{\mathrm{true}}$ as comparator while $\mathrm{Regret}_\pi(K; d^{\pi_\mathrm{E}})$ is the pseudo regret in a Linear MDP with adversarial costs $\left\{c^k\right\}_{k=1}^{K}$ and having the expert occupancy measure as a comparator. The third term involves $\widehat{\mathbf{\Phi}^\intercal d^{\pi_\mathrm{E}}}$ which is the empirical estimates of the expert features expectation vector. It be controlled easily via concentration inequalities ( see Lemma 7).

**Imitation Learning via no-regret algorithms.**   The decomposition in Equation (2) suggests that imitation learning algorithm can be designed chaining one algorithm that updates the sequence $w^k$ to make sure that $\mathrm{Regret}_w(K; w_{\mathrm{true}})$ grows sublinearly and a second one that updates the policy sequence to control $\mathrm{Regret}_\pi(K; d^{\pi_\mathrm{E}})$. Controlling $\mathrm{Regret}_w(K; w_{\mathrm{true}})$ can be easily done via projected online gradient descent Zinkevich (2003).

Unfortunately, controlling $\mathrm{Regret}_\pi(K; d^{\pi_\mathrm{E}})$ is way more challenging because we have no knowledge of the transition dynamics. Therefore, we can not project on the feasible set of occupancy measures. To circumvent this issue we rely on the recent literature Luo et al. (2021); Sherman et al. (2023b); Dai et al. (2023) that however focuses on bandit feedback. In our case, the $\pi$ player has full information on the cost vector $c^k$. Thus, we design a simpler algorithm Algorithm 1 which achieves a better regret bound in the easier full information case. Algorithm 1 improves over the regret bound in Zhong & Zhang (2023) and easily extends to the infinite horizon setting (see Algorithm 2).

**Improved algorithm for finite horizon**   The techniques explained so far do not allow to get the better bound of order $\mathcal{O}(\sqrt{K})$ in the finite horizon setting (see Equation (1)). The idea is to let the $w$ player update first, then the $\pi$ player can update their policy knowing in advance the loss that they will suffer. This allows to use LSVI-UCB Jin et al. (2019) for the $\pi$-player which has been originally designed for a fixed cost but we show that it still guarantees $\mathcal{O}(\sqrt{T})$ regret against an arbitrary sequence of costs when the learner knows in advance the cost function at the next episode. On the other hand, LSVI-UCB suffers linear regret if the adversarial loss is not known in advance so letting the $w$ player update first is crucial. This result is provided in Appendix A.

## 4   WARM UP: ONLINE LEARNING IN ADVERSARIAL LINEAR MDP

We start by presenting our result in full information episodic linear MDP with adversarial costs that improves over Zhong & Zhang (2023) by a factor $H^{1/2}$. The algorithm is quite simple. We apply a policy iteration like method with two important twist: (i) in the policy improvement step, we update the policy with a no regret algorithm rather than a greedy step. Moreover, the policy is updated only every $\tau$ episodes using as loss vector the average $Q$ value over the last batch of collected episodes, (ii) in the policy evaluation step, we compute an optimistic estimate of the $Q$ function for the current policy using only on-policy data.

The last part is crucial because the use of off-policy data makes the covering argument for Linear MDP problematic. Indeed, one would need to cover the space of stochastic policy when computing the covering number of the value function class but this leads to the undesirable dependence on the number of states and actions for the log covering number (see for example Abbasi-Yadkori et al. (2013)). An alternative bound on the covering number shown in Zhong & Zhang (2023) would instead lead to linear regret.

---

**Algorithm 1** On-policy MDP-E with unknown transitions and adversarial costs.

---

1: **Input:** Dataset size $\tau$, Exploration parameter $\beta$, Step size $\eta$, initialize $\pi_0$ as uniform distribution over $\mathcal{A}$
2: **for** $j = 1, \ldots \lfloor K/\tau \rfloor$ **do**
3:     Denote the indices interval $T_j \triangleq [(j-1) \lfloor K/\tau \rfloor, j \lfloor K/\tau \rfloor)$.
4:     // Collect on-policy data
5:     Collect $\tau$ trajectories with policy $\pi^{(j)}$ and store them in the dataset $\mathcal{D}_h^{(j)} = \left\{ (s_h^i, a_h^i, \mathbf{c}_h^i, s_{h+1}^i) \right\}_{i \in T_j}$.
6:     Denote global dataset $\mathcal{D}^{(j)} = \cup_{h=1}^H \mathcal{D}_h^{(j)}$.
7:     **for** $k \in T_j$ **do**
8:         // Optimistic policy evaluation
9:         Initialize $V_{H+1}^k = 0$
10:        **for** $h = H, \ldots, 1$ **do**
11:            $\Lambda_h^k = \sum_{(s,a) \in \mathcal{D}_h^{(j)}} \phi(s,a)\phi(s,a)^{\mathsf{T}} + I$     // $\phi(s,a)$ is the $(s,a)^{\text{th}}$ row of the matrix $\mathbf{\Phi}$.
12:            $\mathbf{v}_h^k = (\Lambda_h^k)^{-1} \sum_{(s,a,s') \in \mathcal{D}_h^{(j)}} \phi(s,a) V_{h+1}^k(s')$
13:            $b_h^k(s,a) = \beta \|\phi(s,a)\|_{(\Lambda_h^k)^{-1}}$
14:            $Q_h^k = \left[ \mathbf{c}_h^k + \mathbf{\Phi}\mathbf{v}_h^k - b_h^k \right]_0^{H-h+1}$
15:            $V_h^k(s) = \langle \pi_h^k(s), Q_h^k(s, \cdot) \rangle$   (with $\pi^k = \pi^{(j)}$).
16:        **end for**
17:    **end for**
18:    // Policy Improvement Step
19:    Compute average $Q$ value $\bar{Q}_h^{(j)}(s,a) = \frac{1}{\tau} \sum_{k \in T_j} Q_h^k(s,a)$.
20:    Update policy $\pi_h^{(j+1)}(a|s) \propto \exp\left( -\eta \sum_{i=1}^j \bar{Q}_h^{(i)}(s,a) \right)$
21: **end for**

---

Instead, using data collected on-policy allows to apply the covering argument in Sherman et al. (2023b) avoiding the dependence on the number of states and actions. The first twist is at this point necessary to make the policy updates more rare giving the possibility to collect more on-policy episodes with a fixed policy. The algorithm pseudocode is in Algorithm 1.

## 4.1 ANALYSIS

**Theorem 3.** *Under Assumption 1, run Algorithm 1 with exploration parameter $\beta = \tilde{\mathcal{O}}(dH)$, dataset size $\tau = \frac{5\beta}{2}\sqrt{\frac{Kd}{\log|\mathcal{A}|}}$ and step size $\eta = \sqrt{\frac{\tau \log|\mathcal{A}|}{KH^2}}$. Then, it holds with probability $1 - \delta$, that*

$$\text{Regret}(K; \pi^\star) = \sum_{k=1}^K V_1^{\pi^k, k}(s_1) - V_1^{\pi^\star, k}(s_1) \leq \tilde{\mathcal{O}}\left( d^{3/4} H^{3/2} \log^{1/4}|\mathcal{A}| \, K^{3/4} \log \frac{K}{\delta} \right) \quad (3)$$

*where we use the compact notation $V_h^{\pi, k}(\cdot) \triangleq V_h^\pi(\cdot; c^k)$.*

*Proof. Sketch* Adding and subtracting the term $\sum_{k=1}^K V_1^k(s_1)$ in the definition of regret, we have that defining $\delta_h^k(s,a) \triangleq \mathbf{c}_h^k(s,a) + P_h V_{h+1}^k(s,a) - Q_h^k(s,a)$

$$\text{Regret}(K; \pi^\star) = \sum_{k=1}^K V_1^{\pi^k, k}(s_1) - V_1^k(s_1) + V_1^k(s_1) - V_1^{\pi^\star, k}(s_1)$$

$$\leq \sum_{k=1}^K \sum_{h=1}^H \mathbb{E}_{s \sim d_h^{\pi^\star}} \left[ \langle Q_h^k(s, \cdot), \pi_h^k(s) - \pi_h^\star(s) \rangle \right] - \sum_{k=1}^K \sum_{h=1}^H \mathbb{E}_{s,a \sim d_h^{\pi^\star}} \left[ \delta_h^k(s,a) \right]$$

$$+ \sum_{k=1}^K \sum_{h=1}^H \mathbb{E}_{s,a \sim d_h^{\pi^k}} \left[ \delta_h^k(s,a) \right]$$

where the last inequality holds by the extended performance difference lemma Cai et al. (2020); Shani et al. (2020). At this point we can invoke Lemma 2 ( see Appendix E) to obtain

$$-2b_h^k(s,a) \le Q_h^k(s,a) - \mathbf{c}_h^k(s,a) - P_h V_{h+1}^k(s,a) \le 0 \quad \forall (s,a) \in \mathcal{S} \times \mathcal{A}, h \in [H], k \in [K] \quad (4)$$

with probability $1 - \delta$. This implies that with probability $1 - \delta$

$$\mathrm{Regret}(K; \pi^\star) \le \sum_{k=1}^{K} \sum_{h=1}^{H} \mathbb{E}_{s \sim d_h^{\pi^\star}} \left[ \langle Q_h^k(s,\cdot), \pi_h^k(s) - \pi_h^\star(s) \rangle \right] + 2 \sum_{k=1}^{K} \sum_{h=1}^{H} \mathbb{E}_{s,a \sim d_h^{\pi^k}} \left[ b_h^k(s,a) \right]$$

$$\le \frac{\tau \log |\mathcal{A}|}{\eta} + \tau H + \eta K H^2 + 2 \sum_{k=1}^{K} \sum_{h=1}^{H} \mathbb{E}_{s,a \sim d_h^{\pi^k}} \left[ b_h^k(s,a) \right]$$

Notice that the last inequality follows from the mirror descent with blocking result (Sherman et al., 2023b, Lemma F.5). Then, by (Sherman et al., 2023b, Lemma C.5), it holds that with probability $1 - 2\delta$.

$$\mathrm{Regret}(K; \pi^\star) \le \frac{\tau \log |\mathcal{A}|}{\eta} + \tau H + \eta K H^2 + \frac{10 K H \sqrt{d} \beta \log(2\tau/\delta)}{\sqrt{\tau}}$$

The proof is concluded plugging in the values specified in the theorem statement. $\qquad \square$

## 4.2 EXTENSION TO THE INFINITE HORIZON SETTING.

We show our proposed extension to the infinite horizon in Algorithm 2. The main difference in the analysis is to the handle the fact that in the infinite horizon setting we can not run a backward recursion to compute the optimistic value functions as done in Steps 10-16 of Algorithm 1. Instead, we use the optimistic estimate at the previous iterate $Q^k$ to build an approximate optimistic estimate at the next iterate (see Steps 10-12 in Algorithm 2). The error introduced in this way can be controlled thanks to the regularization in the policy improvement step as noticed in Moulin & Neu (2023). [2]

**Theorem 4.** *Under Assumption 2, consider $K$ iterations of Algorithm 2 run with $\tau \le \frac{K}{\sqrt{\tau}}$ and $\beta = \widetilde{\mathcal{O}}(dH)$, then it holds for any comparator policy $\pi^\star$ that $(1 - \gamma)\mathrm{Regret}(K; \pi^\star) \triangleq \sum_{k=1}^{K} \left\langle d^{\pi^k} - d^{\pi^\star}, \mathbf{c}^k \right\rangle$ is upper bounded with probability $1 - 2\delta$ by*

$$\frac{\tau \log |\mathcal{A}|}{\eta} + \frac{\tau + 1}{1 - \gamma} + \frac{\eta K}{(1 - \gamma)^2} + 12\beta K \sqrt{\frac{d}{\tau}} \log \left( \frac{2K}{\tau \delta} \right) + \frac{\sqrt{2\eta}K}{(1 - \gamma)^2 \tau}.$$

*Proof. Sketch* The proof is based on the following decomposition that holds in virtue of Lemma 1. Denoting $\delta^k(s,a) \triangleq \mathbf{c}^k(s,a) + \gamma P V^k(s,a) - Q^{k+1}(s,a)$ and $g^k(s,a) \triangleq Q^{k+1}(s,a) - Q^k(s,a)$

$$(1 - \gamma)\mathrm{Regret}(K; \pi^\star) = \sum_{k=1}^{K} \mathbb{E}_{s \sim d^{\pi^\star}} \left[ \langle Q^k(s,\cdot), \pi^k(s) - \pi^\star(s) \rangle \right] \qquad \text{(OMD)}$$

$$+ \sum_{k=1}^{K} \sum_{s,a} \left[ d^{\pi^k}(s,a) - d^{\pi^\star}(s,a) \right] \cdot \left[ \delta^k(s,a) \right] \qquad \text{(Optimism)}$$

$$+ \sum_{k=1}^{K} \mathbb{E}_{s,a \sim d^{\pi^k}} \left[ g^k(s,a) \right] - \sum_{k=1}^{K} \mathbb{E}_{s,a \sim d^{\pi^\star}} \left[ g^k(s,a) \right] \qquad \text{(Shift)}$$

Then, we have that Optimism can be bounded similarly to the finite horizon case using Lemma 3 while for Shift we rely on the regularization of the policy improvement step and on the fact that the policy is updated only every $\tau$ steps. All in all, we have that the first term in (Shift) can be bounded as $\frac{1}{(1-\gamma)^2} \sum_{j=2}^{\lfloor K/\tau \rfloor} \sqrt{2\eta} = \frac{\sqrt{2\eta}K}{(1-\gamma)^2 \tau}$. The second term just telescopes therefore (Shift) $\le \frac{\sqrt{2\eta}K}{(1-\gamma)^2 \tau} + \frac{1}{1-\gamma}$. Finally, (OMD) can be bounded as in the finite horizon case. $\qquad \square$

---

[2]Regularization in both the evaluation and improvement step has been proven successful in the infinite horizon linear mixture MDP setting Moulin & Neu (2023). Their regularization in the evaluation step is helpful to improve the horizon dependence. In our case, we use unregularized evaluation because in the linear MDP setting our analysis presents additional leading terms that can not be bounded as $\mathcal{O}(\sqrt{K})$ with regularization in the policy evaluation routine.

---

**Algorithm 2** Infinite Horizon Linear MDP with adversarial losses.

---

1: **Input:** Dataset size $\tau$, Exploration parameter $\beta$, Step size $\eta$, Initial policy $\pi_0$ (uniform over $\mathcal{A}$), initialize $V^1 = 0$.
2: **for** $j = 1, \ldots \lfloor K/\tau \rfloor$ **do**
3:     // Collect on-policy data
4:     Denote the indices interval $T_j \triangleq [(j-1)\lfloor K/\tau \rfloor, j\lfloor K/\tau \rfloor)$.
5:     Sample $\mathcal{D}^{(j)} = \left\{ s^i, a^i, s'^{,i}, \mathbf{c}^i(s^i) \right\}_{i \in T_j} \sim d^{\pi^{(j)}}$ using (Agarwal et al., 2020b, Algorithm 1).
6:     Compute $\Lambda^{(j)} = \sum_{(s,a) \in \mathcal{D}^{(j)}} \phi(s,a)\phi(s,a)^\intercal + I$.
7:     Compute $b^{(j)}(s,a) = \beta \|\phi(s,a)\|_{(\Lambda^{(j)})^{-1}}$.
8:     // Optimistic Policy Evaluation
9:     **for** $k \in T_j$ **do**
10:        $\mathbf{v}^k = (\Lambda^{(j)})^{-1} \sum_{(s,a,s') \in \mathcal{D}^{(j)}} \phi(s,a)V^k(s')$
11:        $Q^{k+1} = \left[ \mathbf{c}^k + \gamma \mathbf{\Phi} \mathbf{v}^k - b^{(j)} \right]_0^{(1-\gamma)^{-1}}$
12:        $V^{k+1}(s) = \left\langle \pi^{(j)}(a|s), Q^{k+1}(s,a) \right\rangle$
13:     **end for**
14:     // Policy Improvement Step
15:     Compute average $Q$ value $\bar{Q}^{(j)}(s,a) = \frac{1}{\tau} \sum_{k \in T_j} Q^k(s,a)$.
16:     Update policy: $\pi^{(j+1)}(a|s) \propto \exp\left( -\eta \sum_{i=1}^{j} \bar{Q}^{(i)}(s,a) \right)$
17: **end for**

---

## 5   IMITATION LEARNING IN INFINITE HORIZON MDPS

In this section, we apply Theorem 4 to imitation learning. Indeed, we design Algorithm 3 using the insights from the decomposition in Equation (2): we use a no regret algorithm to update the cost at each round and we update the learner's policy using a no regret algorithm for infinite horizon full information adversarial Linear MDP, of which Algorithm 2 is the first example in the literature. The

---

**Algorithm 3** Imitation Learning via Adversarial Reinforcement Learning (**ILARL**) for Infinite Horizon Linear MDPs.

---

1: **Input:** Access to Algorithm 2 (with inputs $\tau, \beta, \eta, \pi_0$), Step size for Cost Update $\alpha$, Expert dataset $\mathcal{D}_{\pi_E} = \{\boldsymbol{\tau}_i\}_{i=1}^{\tau_E}$.
2: Estimate for expert feature visitation $\widehat{\mathbf{\Phi}^\intercal d^{\pi_E}} \triangleq \frac{(1-\gamma)}{\tau_E} \sum_{\boldsymbol{\tau} \in \mathcal{D}_{\pi_E}} \sum_{s_h, a_h \in \boldsymbol{\tau}} \gamma^h \phi(s_h, a_h)$.
3: **for** $k = 1, \ldots, K$ **do**
4:     // Cost Update (to control $\text{Regret}_w(K; w_{\text{true}})$)
5:     Estimate $\widehat{\mathbf{\Phi}^\intercal d^{\pi^k}} \triangleq \tau^{-1} \sum_{s,a \in \mathcal{D}^{(j)}} \phi(s,a)$ where $\mathcal{D}^{(j)}$ is defined in Step 6 in Algorithm 2.
6:     $\mathbf{w}^{k+1} = \Pi_\mathcal{W} \left[ \mathbf{w}^k - \alpha(\widehat{\mathbf{\Phi}^\intercal d^{\pi_E}} - \widehat{\mathbf{\Phi}^\intercal d^{\pi^k}}) \right]$ with $\mathcal{W} = \{\mathbf{w} : \|\mathbf{w}\|_2 \leq 1\}$.
7:     // Policy Update ( to control $\text{Regret}_\pi(K; \pi_E)$ )
8:     The cost $(\mathbf{\Phi}\mathbf{w}^k + 1)/2$ is revealed to the learner.
9:     The learner updates their policy $\pi^k$ performing one iteration of Algorithm 2.
10: **end for**

---

guarantees for Algorithm 3 are given in the following theorem.

**Theorem 5.** *Under Assumptions 2,3, let us consider $K$ iterations of Algorithm 3 with $K \geq \widetilde{\mathcal{O}}\left( \frac{\log|\mathcal{A}|d\beta^2 \log^2(1/\delta)}{(1-\gamma)^6 \epsilon^4} \right)$ where $\beta$ is chosen as in Lemma 5 (i.e. $\beta = \widetilde{\mathcal{O}}(d(1-\gamma))^{-1}$). Moreover, let consider the following choices $\alpha = \frac{1}{\sqrt{2K}}$, $\tau = \mathcal{O}\left( \frac{\beta(1-\gamma)\sqrt{dK}\log(2dK/\delta)}{\sqrt{\log|\mathcal{A}|}} \right)$, expert trajectories $\tau_E = \frac{8d\log(d/\delta)}{(1-\gamma)^2 \epsilon_E^2}$ and $\eta = \sqrt{\frac{\tau \log|\mathcal{A}|(1-\gamma)^2}{K}}$. Then, the above conditions ensure $\frac{1}{1-\gamma} \left\langle \mathbf{c}_{\text{true}}, d^{\pi_E} - \frac{1}{K}\sum_{k=1}^{K} d^{\pi^k} \right\rangle \leq \epsilon + \epsilon_E$ with probability $1 - 4\delta$.*

The proof included in Appendix E starts with the decomposition in Equation (2). Then, we control the term $\text{Regret}_w(K; w_{\text{true}})$ with the standard online gradient descent analyses and the term $\text{Regret}_\pi(K; \pi_E)$ with Theorem 4. Finally, we control the statistical estimation error with an application of Lemma 7.

**Remark 2.** *The resulting algorithm improves over Viano et al. (2022) in two ways: (i) We bypass all kind of exploration assumptions, such as the persistent excitation assumption. We remark that this a qualitative improvement. Indeed, the persistent excitation assumption is easily violated by deterministic policies with tabular features. (ii) Moreover, the sample complexity improves from $\mathcal{O}(\epsilon^{-5})$ to $\mathcal{O}(\epsilon^{-4})$.*

## 6   EMPIRICAL EVALUATION

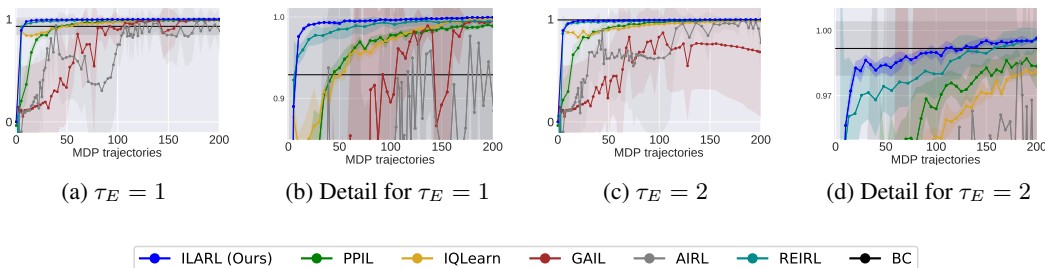

(a) $\tau_E = 1$  (b) Detail for $\tau_E = 1$  (c) $\tau_E = 2$  (d) Detail for $\tau_E = 2$

ILARL (Ours) — PPIL — IQLearn — GAIL — AIRL — REIRL — BC

Figure 1: Experiments on a continuous gridworld with a stochastic expert. The $y$-axis reports the normalized return. 1 correpsonds to the expert performance and 0 to the uniform policy one.

We numerically verify the main theoretical insights derived in the previous sections (i) We aim to verify that for a general stochastic expert, the efficiency in terms of expert trajectories improves upon behavioural cloning. (ii) ILARL is more efficient in terms of MDP trajectories compared to PPIL Viano et al. (2022) which has worst theoretical guarantees and with popular algorithms that are widely used in practice but do not enjoy theoretical guarantees: GAIL Ho et al. (2016), AIRL Fu et al. (2018), REIRL Boularias et al. (2011) and IQLearn Garg et al. (2021) The experiments are run in a continuous state MDP explained in Appendix G.

**Expert trajectory efficiency with stochastic expert** For the first claim, we use a stochastic expert obtained following with equal probability either the action taken by a deterministic experts previously trained with LSVI-UCB or an action sampled uniformly at random. We collect with such policy $\tau_E$ trajectories. From Figure 1, we observe that all imitation learning we tried have a final performance improving over behavioural cloning for the case $\tau_E = 1$ while only REIRL and ILARL do so for $\tau_E = 2$. In both cases, ILARL achieves the highest return that even matches the expert performance.

**MDP trajectories efficiency** For the second claim, we can see in Figure 1 that ILARL is the most efficient algorithm in terms of MDP trajectories for both values of $\tau_E$.

## 7   CONCLUSIONS

In this paper, we proposes ILARL which greatly reduces the number of MDP trajectories in imitation learning in Linear MDP and BRIG that provides a further improvement for the finite horizon case. Both results build on the connection between imitation learning and MDPs with adversarial losses.

There is a number of exciting future directions. In particular, the estimation of $\widehat{\Phi^\top d^{\pi_E}}$ could be carried out with **fewer expert trajectories** using trajectory access to the MDP. This observation has been proven successful having access to the exact transitions of the MDP in the tabular case Rajaraman et al. (2020) or under linear function approximation with further assumption on the expert policy and the feature distribution Swamy et al. (2022); Rajaraman et al. (2021). Whether the same is possible for general stochastic experts in Linear MDP is an interesting open question. Finally, a **better sample complexity** can be achieved designing better no regret algorithm for infinite horizon adversarial discounted linear MDP with full-information feedback and apply them in Step 9 of Algorithm 3 building for example on the recent result for the finite horizon case in Sherman et al. (2023a).

**Reproducibility statement** The experimental details are provided in Appendix G and in the README file of the attached code. For the theoretical part, the assumptions are clearly stated in Section 2.1.

**Ethics Statement** The authors acknowledge that they have read and adhere to the ICLR Code of Ethics.

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

# A  AN IMPROVEMENT FOR THE FINITE HORIZON CASE

We notice that in Algorithm 3 we missed an opportunity. In fact, we could use prior knowledge of the cost function $\mathbf{c}_h^k$ to update the policy $\pi_h^k$. In other words, in Step 8 of Algorithm 3 we could reveal $\mathbf{w}^{k+1}$ to the algorithm that controls $\text{Regret}_\pi(K; d^{\pi_E})$. From an online learning perspective we can play best response to control more effectively the regret term $\text{Regret}_\pi(K; d^{\pi_E})$ using LSVI-UCB Jin et al. (2019). This idea leads to the Algorithm 4. Theorem 6 proves that Algorithm 4 improves

---

**Algorithm 4** Best Response Imitation learninG (**BRIG**).

1: **Input:** Exploration parameter $\beta$, Step size $\eta$ , Reward step size $\alpha$, expert dataset $\mathcal{D}_{\pi_E} = \cup_{h \in [H]} \mathcal{D}_{\pi_E, h}$.
2: Initialize $\pi_0$ as uniform distribution over $\mathcal{A}$.
3: Estimate expert features expectation vectors $\widehat{\mathbf{\Phi}^\intercal d_h^{\pi_E}} = \frac{1}{\left| \mathcal{D}_{\pi_E, h} \right|} \sum_{s, a \in \mathcal{D}_{\pi_E, h}} \phi(s, a)$ for all $h \in [H]$.
4: **for** $k = 1, \ldots K$ **do**
5:     Collect one episodes with policy $\pi^k$ denoted as $\boldsymbol{\tau}^k = \left\{ (s_h^k, a_h^k, s_{h+1}^k) \right\}_{h=1}^{H-1}$ and for every $h \in [H]$ append data $(s_h^k, a_h^k, s_{h+1}^k)$ to $\mathcal{D}_h$.
6:     // Cost Update (to control $\text{Regret}_w(K; w_{\text{true}})$)
7:     $\mathbf{w}_h^{k+1} = \Pi_{\mathcal{W}} \left[ \mathbf{w}_h^k - \alpha (\widehat{\mathbf{\Phi}^\intercal d_h^{\pi_E}} - \phi(s_h^k, a_h^k)) \right]$ with $\mathcal{W} = \{ \mathbf{w} : 0 \leq \mathbf{w} \leq 1 \}$ for all $h \in [H]$.
8:     // Full information LSVI-UCB ( to control $\text{Regret}_\pi(K; \pi_E)$)
9:     Initialize $V_{H+1}^k = 0$
10:    **for** $h = H, \ldots, 1$ **do**
11:        $\Lambda_h^k = \sum_{(s,a) \in \mathcal{D}_h} \phi(s, a) \phi(s, a)^\intercal + I$
12:        $\mathbf{v}_h^k = (\Lambda_h^k)^{-1} \sum_{(s, a, s') \in \mathcal{D}_h} \phi(s, a) V_{h+1}^k(s')$
13:        $b_h^k(s, a) = \beta \left\| \phi(s, a) \right\|_{(\Lambda_h^k)^{-1}}$
14:        $Q_h^k = \left[ \mathbf{\Phi} \mathbf{w}_h^{k+1} + \mathbf{\Phi} \mathbf{v}_h^k - b_h^k \right]_0^{H-h+1}$     // We use the future loss $\mathbf{w}_h^{k+1}$.
15:        $\pi_h^{k+1}(s) = \arg \min \left( Q_h^k(s, \cdot) \right)$     // Greedy policy update, Best Response.
16:        $V_h^k(s) = \left\langle \pi_h^{k+1}(s), Q_h^k(s, \cdot) \right\rangle$.
17:    **end for**
18: **end for**

---

the required number of interaction to $K = \widetilde{\mathcal{O}}(H^4 d^3 \epsilon^{-2} \log^2(1/\delta))$ which greatly improves over $K = \widetilde{\mathcal{O}}(H^6 \log |\mathcal{A}| \, d^3 \epsilon^{-4} \log^2(1/\delta))$ achieved by Algorithm 3 applied to finite horizon problems which does not use the best response observation. A core step in the proof is to show that the regret of LSVI-UCB is still $\mathcal{O}(\sqrt{K})$ if the cost function is not fixed but it is observed in advanced by the agent.

**Theorem 6.** *Let us consider* $K = \mathcal{O}\left( \frac{H^4 d^3 \log(dH/(\epsilon \delta))}{\epsilon^2} \right)$ *iterations of Algorithm 4 run with* $\alpha = \sqrt{\frac{1}{2K}}$ *and expert demonstrations* $\tau_E = \frac{2H^2 d \log(2d/\delta)}{\epsilon_E^2}$. *Moreover, let* $\hat{k}$ *be an iteration index sampled uniformly at random from* $\{1, 2, \ldots, K\}$, *then it holds that with probability* $1 - 3\delta$, *it holds that* $\mathbb{E}_{\hat{k}} \left[ V_1^{\pi^{\hat{k}}}(s_1; \mathbf{c}_{\text{true}}) - V_1^{\pi_E}(s_1; \mathbf{c}_{\text{true}}) \right] \leq \mathcal{O}(\epsilon_E + \epsilon).$

**Remark 3.** *Unfortunately, the best response idea does not help improving the infinite horizon result because the use of greedy policies makes the term* (Shift) *impossible to control.*

# B  INTERACTION PROTOCOL

---

**Protocol 1** Interaction in Adversarial MDPs

---

1: **for** Episode index $k \in [1, K]$ **do**
2:    Sample initial state $s_1^k \sim \boldsymbol{\nu}_0$
3:    **if** Finite Horizon **then**
4:       **for** stage $h \in [0, H]$ **do**
5:          The learner plays an action sampled from the policy $a_h^k \sim \pi_h^k(\cdot|s_h^k)$.
6:          The environment sample next state $s_h^k \sim P_h(\cdot|s_h^k, a_h^k)$.
7:          The agent observes the vector $c_h^k$.
8:       **end for**
9:    **end if**
10:    **if** Infinite Horizon **then**
11:       Initialize $Z = 0$, $i = 1$, $s^1 \sim \boldsymbol{\nu}_0$.
12:       **while** $Z == 0$ **do**
13:          The learner plays an action sampled from the policy $a^i \sim \pi^k(\cdot|s^i)$.
14:          The environment sample next state $s'^{,i} \sim P(\cdot|s^i, a^i)$, $s^{i+1} = s'^{,i}$.
15:          The agent observes the vector $c^k$.
16:          // Restart with probability $1 - \delta$.
17:          Sample $Z \sim \text{Bernoulli}(1 - \gamma)$.
18:       **end while**
19:    **end if**
20:    The learner chooses her next policy, i.e. $\pi^{k+1}$.
21: **end for**

---

## C   Future directions

**Reducing the number of expert trajectories**   Rajaraman et al. (2021) showed that under known transitions for a particular choice of features and when the linear expert occupancy measure is uniform over the state space the necessary expert trajectories are $\mathcal{O}\left(\frac{H^{3/2}d^2}{\epsilon}\right)$.

In the linear MDP case, under the same assumption on the features we can show that the same amount of expert trajectories is sufficient even for the unknown transition case. Moreover, the algorithm we propose is computationally efficient while it is unclear how the output policy in linear Mimic-MD Rajaraman et al. (2021) can be computed with complexity independent on the number of states and actions. We detail this in Appendix H.

In addition, it is an interesting open question to see if the same improvement in terms of expert trajectory can be achieved under weaker conditions. A first step has been already made in Swamy et al. (2022). They still requires a linear expert model but avoids the uniform expert occupancy measure assumption replacing it with the bounded density assumption (see (Swamy et al., 2022, Assumption 9)). A natural follow up would be to investigate if the same expert trajectory bound can be obtained bringing the Linear MDP assumption but dropping the linear expert and the other assumptions used in Rajaraman et al. (2021); Swamy et al. (2022). An intermediate step could benefit from using the persistent excitation assumption which is the natural counterpart of the bounded density assumption (Swamy et al., 2022, Assumption 9) in Linear MDPs.

**Results for Linear Mixture MDPs**   Analogous ideas can be used for the case of Linear Mixture MDPs. In particular, one can use the same structure as in Algorithm 3 but replacing an algorithm that deals with adversarial losses in Linear Mixture MDPs. For the infinite horizon case one can use Moulin & Neu (2023) while for the finite horizon case one can choose Zhou et al. (2021). In the latter case one can improve the Sample Complexity of OGAIL Liu et al. (2022) to $\mathcal{O}\left(H^3 d^2 \epsilon^{-2}\right)$. Moreover, we do not think that that the Linear Expert assumption Rajaraman et al. (2021) is meaningful in Linear Mixture MDPs because this would require the learner to know in advance the features $\int_{\mathcal{S}} \phi(s, a, s')V^\star(s')ds'$ where $V^\star$ is the optimal state value function.

**Extension to Bilinear Classes**   The current results can be extended to Bilinear Classes Du et al. (2021) at least in the finite horizon case. To this end we would need the observation that we can update the cost first and then using an algorithm which is allowed to see the next cost vector one

round in advance. This is the same fact that allowed us to obtain $\mathcal{O}(\epsilon^{-2})$ sample complexity bound in the finite horizon case using LSVI-UCB.

In the following we present an informal discussion of the proof technique that would prove polynomial sample complexity for Imitation Learning in bilinear classes.

If the cost at round $k$ is known before the learner needs to take an action the agent can form the discrepancy function

$$\ell^k(s_h, a_h, s_{h+1}, g) = Q_{h,g}(s_h, a_h) + c_h^k - V_{h+1,g}(s_{h+1})$$

where the upper script $k$ highlights the fact that the discrepancy function depends on the adversarial cost $c^k$. At each round, we can then compute

$$\text{argmax}_{g \in \mathcal{H}} V_{0,g}(s_1) \quad \text{s.t.} \quad \frac{1}{m} \sum_{\tau=1}^{k} \sum_{i=1}^{m} \ell^k(s_h^i, a_h^i, s_{h+1}^i, g) \le k\epsilon_{\text{gen},k}^2(m, \delta) \quad \forall h \in [H]$$

Where the generalization error at round $k$ denoted $\epsilon_{\text{gen},k}$ satisfies that

$$\sup_{g \in \mathcal{H}} \left| \frac{1}{m} \sum_{i=1}^{m} \ell^k(s_h^i, a_h^i, s_{h+1}^i, g) - \mathbb{E}_{s,a \sim d^{\pi^k}, s' \sim P(\cdot|s,a)} \left[ \ell^k(s_h, a_h, s_{h+1}, g) \right] \right| \le \epsilon_{\text{gen},k}(m, \delta)$$

with probability at least $1 - \delta$.

At this point, we modify the Bilinear Classes assumption to keep into account the adversarial costs setting as follows. We assume that all the adversarial costs belongs to a convex set $\mathcal{C}$. Then we consider an MDP for which there exists a function $f^\star \in \mathcal{H}$ such that for all $c \in \mathcal{C}$, it holds that

$$\left| \mathbb{E}_{s,a \sim d_h^{\pi_f}} \left[ Q_{h,f}(s_h, a_h) + c_h(s_h, a_h) - V_{h+1,f} \right] \right| \le \left| \langle W_h^c(f) - W_h^c(f^\star), X_h^c(f) \rangle \right|$$

and

$$\left| \mathbb{E}_{s,a \sim d_h^{\pi_f}} \left[ Q_{h,g}(s_h, a_h) + c_h(s_h, a_h) - V_{h+1,g} \right] \right| \le \left| \langle W_h^c(g) - W_h^c(f^\star), X_h^c(f) \rangle \right|$$

This modified assumption for Bilinear classes with time changing rewards implies that the comparator hypothesis $f^\star$ is realizable for all $k$, that is, for all $k$ it holds that

$$\frac{1}{m} \sum_{\tau=1}^{k} \sum_{i=1}^{m} \ell^k(s_h^i, a_h^i, s_{h+1}^i, f^\star) \le k\epsilon_{\text{gen},k}^2(m, \delta)$$

so optimism holds and with the same steps in Du et al. (2021) it can be proven that:

$$\text{Regret}(K, \pi^\star) \le \sum_{k=1}^{K} \left| \mathbb{E}_{s,a \sim d_h^{\pi_{f^k}}} \left[ Q_{h,f^k}(s_h, a_h) + c_h(s_h, a_h) - V_{h+1,f^k} \right] \right|$$

$$\le \sum_{k=1}^{K} \left| \left\langle W_h^{r^k}(f^k) - W_h^{r^k}(f^\star), X_h^{r^k}(f^k) \right\rangle \right|$$

Then, using (Du et al., 2021, Equation 8) and the elliptical potential lemma we obtain

$$\frac{1}{K} \text{Regret}(K, \pi^\star) = \mathcal{O} \left( \sqrt{K \max_{k \in [K]} \epsilon_{\text{gen},k}^2(m, \delta) \left( \exp \frac{\log K}{K} - 1 \right)} \right)$$

Then, denoting $\epsilon_{\text{gen}}(m, \delta) = \max_{k \in [K]} \epsilon_{\text{gen},k}(m, \delta)$ and choosing $K = \mathcal{O}(\log \epsilon_{\text{gen}}^{-2}(m, \delta))$ gives $\frac{1}{K} \text{Regret}(K, \pi^\star) \le \mathcal{O}(\epsilon_{\text{gen}}(m, \delta))$.

This concludes the regret proof for the $\pi$-player. The player updating the sequences of cost can still use OGD projecting on the set $\mathcal{C}$.

Our algorithm for the finite horizon setting, BRIG, uses greedy policies. In this case, Du et al. (2021) showed that the generalization error can be controlled effectively in many instances such as Linear $Q^\star/V^\star$, Low Occupancy measure models Du et al. (2021), Bellman complete models Munos (2003) and finite Bellman Rank Jiang et al. (2017).

However in the infinite horizon case we need to use regularization for which it is currently not known if $\mathrm{gen}(m, \delta)$ can be controlled effectively. This is again related to the issue with the covering number of softmax policies in linear MDPs ( see Sherman et al. (2023a); Zhong & Zhang (2023) ). This is an interesting open question.

A final comment is that the algorithm proposed in Du et al. (2021) for bilinear classes is not computationally efficient is general. Therefore also its adversarial extension presented above will have this drawback. In this paper we focused on the smaller class of Linear MDP for which we provide a computationally efficient algorithm.

**Improving dependence on $\epsilon$, $d$ and $H$**    For what concerns the finite horizon case we presented BRIG which has a dependence on $\epsilon$ which can not be improved further if not bypassing the reduction to online learning in MDPs. However the dependence in $d$ and $H$ can be improved. To circumvent this problem, we think that one could apply LSVI-UCB++ He et al. (2023). Indeed we think that as LSVI-UCB, also LSVI-UCB++ still enjoys regret guarantees if the cost changes adversarially but the learner knows the cost at the next round. In this case the Sample Complexity of Algorithm 4 could be improved by a factor $dH$, i.e. to $\widetilde{\mathcal{O}}\left(d^2 H^3 \epsilon^{-2}\right)$. For sake of simplicity, we leave this improvement for future works.

# D    OMITTED PROOFS

## D.1    PROOF OF LEMMA 1

**Lemma 1.** *Consider the MDP $M = (\mathcal{S}, \mathcal{A}, \gamma, P, \mathbf{c})$ and two policies $\pi, \pi' : \mathcal{S} \to \Delta_\mathcal{A}$. Then consider for any $\widehat{Q} \in \mathbb{R}^{|\mathcal{S}||\mathcal{A}|}$ and $\widehat{V}^\pi(s) = \left\langle \pi(\cdot|s), \widehat{Q}(s, \cdot) \right\rangle$ and $Q^{\pi'}, V^{\pi'}$ be respectively the state action and state value function of the policy $\pi$ in MDP $M$. Then, it holds that*

$$\left\langle \boldsymbol{\nu}_0, \widehat{V}^\pi - V^{\pi'} \right\rangle = \frac{1}{1-\gamma}\left( \left\langle d^{\pi'}, \widehat{Q}^\pi - \mathbf{c} - \gamma P\widehat{V}^\pi \right\rangle + \left\langle d^{\pi'}, E\widehat{V}^\pi - \widehat{Q}^\pi \right\rangle \right)$$

*Proof.* Consider the Bellman equation in vector form, i.e. $Q^{\pi'} = \mathbf{c} + \gamma P V^{\pi'}$. Then, let us add and subtract the term $\mathbf{c} + \gamma P\widehat{V}^\pi$ and let us consider on both sides the inner product with the occupancy measure $d^{\pi'}$.

$$\left\langle d^{\pi'}, \widehat{Q}^\pi \right\rangle = \left\langle d^{\pi'}, \widehat{Q}^\pi - \mathbf{c} - \gamma P\widehat{V}^\pi \right\rangle + \left\langle d^{\pi'}, \mathbf{c} + \gamma P\widehat{V}^\pi \right\rangle$$
$$= \left\langle d^{\pi'}, \widehat{Q}^\pi - \mathbf{c} - \gamma P\widehat{V}^\pi \right\rangle + (1-\gamma)\left\langle \boldsymbol{\nu}_0, V^{\pi'} \right\rangle + \left\langle \gamma P^\mathsf{T} d^{\pi'}, \widehat{V}^\pi \right\rangle$$
$$= \left\langle d^{\pi'}, \widehat{Q}^\pi - \mathbf{c} - \gamma P\widehat{V}^\pi \right\rangle + (1-\gamma)\left\langle \boldsymbol{\nu}_0, V^{\pi'} - \widehat{V}^\pi \right\rangle + \left\langle E^\mathsf{T} d^{\pi'}, \widehat{V}^\pi \right\rangle$$

Rearranging the terms leads to the conclusion.    □

### D.2 PROOF OF THEOREM 4

*Proof.* Let $V^{\pi^k, k} \in [0, (1-\gamma)^{-1}]^{|\mathcal{S}|}$ denote $V^{\pi^k}(\cdot; \mathbf{c}^k)$, then

$$
\begin{aligned}
\text{Regret}(K; \pi^\star) &\triangleq \sum_{k=1}^{K} \left\langle d^{\pi^k} - d^{\pi^\star}, \mathbf{c}^k \right\rangle \\
&= (1-\gamma) \sum_{k=1}^{K} \left\langle \boldsymbol{\nu}_0, V^{\pi^k, k} - V^{\pi^\star} \right\rangle \\
&= \sum_{k=1}^{K} \mathbb{E}_{s \sim d^{\pi^\star}} \left[ \left\langle Q^k(s, \cdot), \pi^k(s) - \pi^\star(s) \right\rangle \right] \quad \text{(FTRL)} \\
&\quad + \sum_{k=1}^{K} \mathbb{E}_{s, a \sim d^{\pi^\star}} \left[ Q^{k+1}(s, a) - \mathbf{c}^k(s, a) - \gamma P V^k(s, a) \right] \quad \text{(Optimism 1)} \\
&\quad + \sum_{k=1}^{K} \mathbb{E}_{s, a \sim d^{\pi^k}} \left[ \mathbf{c}^k(s, a) + \gamma P V^k(s, a) - Q^{k+1}(s, a) \right] \quad \text{(Optimism 2)} \\
&\quad + \sum_{k=1}^{K} \mathbb{E}_{s, a \sim d^{\pi^k}} \left[ Q^{k+1}(s, a) - Q^k(s, a) \right] \quad \text{(Shift 1)} \\
&\quad + \sum_{k=1}^{K} \mathbb{E}_{s, a \sim d^{\pi^\star}} \left[ Q^k(s, a) - Q^{k+1}(s, a) \right] \quad \text{(Shift 2)}
\end{aligned}
$$

Then, we have that Optimism 1, Optimism 2 can be bounded using Lemma 3 while for Shift 1 we crucially rely on the regularization of the policy improvement step and on the fact that the policy is updated only every $\tau$ steps. Both these observations allow to derive

$$
\begin{aligned}
\sum_{k=1}^{K} \mathbb{E}_{s, a \sim d^{\pi^k}} \left[ Q^k(s, a) - Q^{k+1}(s, a) \right] &\leq \sum_{k=2}^{K} \sum_{s, a} Q^k(s, a) \left( d^{\pi^k}(s, a) - d^{\pi^{k-1}}(s, a) \right) + \sum_{s, a} Q^1(s, a) d^{\pi^1}(s, a) \\
&\leq \sum_{k=2}^{K} \left\| Q^k(s, a) \right\|_\infty \left\| d^{\pi^k} - d^{\pi^{k-1}} \right\|_1 \\
&\leq \frac{1}{1-\gamma} \sum_{k=2}^{K} \left\| d^{\pi^k} - d^{\pi^{k-1}} \right\|_1 \\
&= \frac{1}{1-\gamma} \sum_{j=2}^{\lfloor K/\tau \rfloor} \left\| d^{\pi^{(j)}} - d^{\pi^{(j-1)}} \right\|_1
\end{aligned}
$$

At this point applying Pinkser's inequality and (Moulin & Neu, 2023, Lemma A.1) we obtain

$$
\begin{aligned}
\left\| d^{\pi^{(j)}} - d^{\pi^{(j-1)}} \right\|_1 &\leq \sqrt{2 D_{KL}(d^{\pi^{(j)}}, d^{\pi^{(j-1)}})} \leq \sqrt{\frac{2}{1-\gamma} \mathbb{E}_{s \sim d^{\pi^{(j)}}} D_{KL}(\pi^{(j)}(\cdot|s), \pi^{(j-1)}(\cdot|s))} \\
&= \sqrt{\frac{2\eta}{1-\gamma} \mathbb{E}_{s, a \sim d^{\pi^{(j)}}} \bar{Q}^{(j-1)}} \\
&\leq \frac{\sqrt{2\eta}}{1-\gamma}
\end{aligned}
$$

All in all, we have that

$$
\text{(Shift 1)} \leq \frac{1}{(1-\gamma)^2} \sum_{j=2}^{\lfloor K/\tau \rfloor} \sqrt{2\eta} = \frac{\sqrt{2\eta} K}{(1-\gamma)^2 \tau}
$$

For the second shift term, we can use a trivial telescoping argument to obtain that (Shift 2) $\leq (1-\gamma)^{-1}$.

The terms (Optimism 1) and (Optimism 2) can be bounded exactly as in the finite horizon case thanks to Lemma 3 we have that with probability $1 - \delta$

$$\text{(Optimism 1)} + \text{(Optimism 2)} \le 2 \sum_{k=1}^{K} \mathbb{E}_{s,a \sim d^{\pi^k}} \left[ b^k(s,a) \right]. \tag{5}$$

Therefore, we just need to adapt the argument for bounding the exploration term $2 \sum_{k=1}^{K} \mathbb{E}_{s,a \sim d^{\pi^k}} \left[ b^k(s,a) \right]$ in the infinite horizon case. We start by exploiting the fact that both $b^k$ and $\pi^k$ change in fact only every $\tau$ updates. So we have

$$
\begin{aligned}
\sum_{k=1}^{K} \mathbb{E}_{s,a \sim d^{\pi^k}} \left[ b^k(s,a) \right] &= \tau \sum_{j=1}^{K/\tau} \mathbb{E}_{s,a \sim d^{\pi^{(j)}}} \left[ b^{(j)}(s,a) \right] \\
&= \sum_{j=1}^{K/\tau} \mathbb{E}_{\mathcal{D}^{(j)} \sim d^{\pi^{(j)}}} \left[ \sum_{s,a \in \mathcal{D}^{(j)}} b^{(j)}(s,a) \right] \\
&= \sum_{j=1}^{K/\tau} \mathbb{E}_{\mathcal{D}^{(j)} \sim d^{\pi^{(j)}}} \left[ \sum_{s,a \in \mathcal{D}^{(j)}} \beta \left\| \phi(s,a) \right\|_{(\Lambda^{(j)})^{-1}} \right]
\end{aligned}
$$

At this point, fixing an arbitrary order for the state action pairs in $\mathcal{D}^{(j)}$ we can define for any $i \in [1, \tau]$ the matrix

$$\Lambda_i^{(j)} = \sum_{\ell=1}^{i} \phi(s^\ell, a^\ell) \phi(s^\ell, a^\ell)^T + \lambda I$$

Then, it holds that

$$
\begin{aligned}
\sum_{j=1}^{K/\tau} \mathbb{E}_{\mathcal{D}^{(j)} \sim d^{\pi^{(j)}}} \left[ \sum_{s,a \in \mathcal{D}^{(j)}} \beta \left\| \phi(s,a) \right\|_{(\Lambda^{(j)})^{-1}} \right] &= \sum_{j=1}^{K/\tau} \mathbb{E}_{\mathcal{D}^{(j)} \sim d^{\pi^{(j)}}} \left[ \sum_{\ell=1}^{\tau} \beta \left\| \phi(s^\ell, a^\ell) \right\|_{(\Lambda^{(j)})^{-1}} \right] \\
&\le \sum_{j=1}^{K/\tau} \mathbb{E}_{\mathcal{D}^{(j)} \sim d^{\pi^{(j)}}} \left[ \sum_{\ell=1}^{\tau} \beta \left\| \phi(s^\ell, a^\ell) \right\|_{(\Lambda_\ell^{(j)})^{-1}} \right]
\end{aligned}
$$

At this point, we can notice that for any index pair $\ell, j$ it holds that

$$\left\| \phi(s^\ell, a^\ell) \right\|_{(\Lambda_\ell^{(j)})^{-1}} \le \sqrt{\lambda_{\max}((\Lambda_\ell^{(j)})^{-1})} \left\| \phi(s^\ell, a^\ell) \right\| \le \sqrt{\frac{1}{\lambda_{\min}(\Lambda_\ell^{(j)})}} \le 1$$

where the norm $\phi(s^\ell, a^\ell)$ is upper bounded by 1 thanks to the Linear MDP assumption. Then, via (Sherman et al., 2023b, Lemma F.1) (where the random variable $X_i$ is in this context $\sum_{\ell=1}^{\tau} \beta \left\| \phi(s^\ell, a^\ell) \right\|_{(\Lambda_\ell^{(j)})^{-1}}$ which is supported in $[0, \beta\tau]$) we can continue upper bounding with probability $1 - \delta$ the bonus sum as

$$
\begin{aligned}
\sum_{k=1}^{K} \mathbb{E}_{s,a \sim d^{\pi^k}} \left[ b^k(s,a) \right] &\le 2 \sum_{j=1}^{K/\tau} \sum_{\ell=1}^{\tau} \beta \left\| \phi(s^\ell, a^\ell) \right\|_{(\Lambda_\ell^{(j)})^{-1}} + 4\beta\tau \log(2K/(\tau\delta)) \\
&= 2 \sum_{j=1}^{K/\tau} \sum_{\ell=1}^{\tau} \beta \sqrt{\phi(s^\ell, a^\ell)^T (\Lambda_\ell^{(j)})^{-1} \phi(s^\ell, a^\ell)} + 4\beta\tau \log(2K/(\tau\delta)) \\
&\le 2 \sum_{j=1}^{K/\tau} \beta \sqrt{\tau \sum_{\ell=1}^{\tau} \phi(s^\ell, a^\ell)^T (\Lambda_\ell^{(j)})^{-1} \phi(s^\ell, a^\ell)} + 4\beta\tau \log(2K/(\tau\delta)) \\
&\le 2 \sum_{j=1}^{K/\tau} \beta \sqrt{\tau \sum_{i=1}^{d} \log(1 + \lambda_i)} + 4\beta\tau \log(2K/(\tau\delta))
\end{aligned}
$$

where the last inequality uses (Cesa-Bianchi & Lugosi, 2006, Lemma 11.11) and the notation $\{\lambda_i\}_{i=1}^{d}$ stands for the eigenvalues of the matrix $\Lambda^{(j)} - I$. At this point we can recognize the determinant inside the log and use the determinant trace inequality.

$$
\begin{aligned}
\sum_{j=1}^{K/\tau} \beta \sqrt{\tau \sum_{i=1}^{d} \log\left(1+\lambda_i\right)} &= \sum_{j=1}^{K/\tau} \beta \sqrt{\tau \log\left(\prod_{i=1}^{d} 1 + \lambda_i\right)} \\
&= \sum_{j=1}^{K/\tau} \beta \sqrt{\tau \log\left(\det(\Lambda^{(j)})\right)} \\
&\leq \sum_{j=1}^{K/\tau} \beta \sqrt{\tau d \log\left(\frac{\operatorname{Trace}(\Lambda^{(j)})}{d}\right)} \\
&\leq \sum_{j=1}^{K/\tau} \beta \sqrt{\tau d \log\left(\frac{1 + \operatorname{Trace}(\sum_{\ell=1}^{\tau} \phi(s^\ell, a^\ell)\phi(s^\ell, a^\ell)^T)}{d}\right)} \\
&\leq \sum_{j=1}^{K/\tau} \beta \sqrt{\tau d \log\left(\frac{1 + \tau \max_\ell \operatorname{Trace}(\phi(s^\ell, a^\ell)\phi(s^\ell, a^\ell)^T)}{d}\right)} \\
&= \sum_{j=1}^{K/\tau} \beta \sqrt{\tau d \log\left(\frac{1 + \tau \max_\ell \phi(s^\ell, a^\ell)^T \phi(s^\ell, a^\ell)}{d}\right)} \\
&\leq \sum_{j=1}^{K/\tau} \beta \sqrt{\tau d \log\left(\frac{1+\tau}{d}\right)} \\
&\leq \beta K \sqrt{\frac{d \log 2\tau}{\tau}}.
\end{aligned}
$$

Hence, with probability $1 - 2\delta$ (union bound between the event under which Equation (5) holds and the the application of the concentration result (Sherman et al., 2023b, Lemma F.1) in bounding the exploration bonuses sum), it holds that

$$
\begin{aligned}
\operatorname{Regret}(K; \pi^\star) \leq &\sum_{k=1}^{K} \mathbb{E}_{s \sim d^{\pi^\star}} \left[\langle Q^k(s, \cdot), \pi^k(s) - \pi^\star(s)\rangle\right] + 4\beta K \sqrt{\frac{d \log(2\tau)}{\tau}} + 8\beta \tau \log(2K/(\tau\delta)) \\
&+ \frac{\sqrt{2\eta} K}{(1-\gamma)^2 \tau} + \frac{1}{(1-\gamma)}
\end{aligned}
$$

The last step is to bound (FTRL) invoking (Sherman et al., 2023b, Lemma F.5) noticing that the gradient norm is upper bounded by $\frac{1}{1-\gamma}$. This gives

$$
\sum_{k=1}^{K} \mathbb{E}_{s \sim d^{\pi^\star}} \left[\langle Q^k(s, \cdot), \pi^k(s) - \pi^\star(s)\rangle\right] \leq \frac{\tau \log|\mathcal{A}|}{\eta} + \frac{\tau}{1-\gamma} + \frac{\eta K}{(1-\gamma)^2}
$$

Putting all together, we have that for $\tau \leq \frac{K}{\sqrt{\tau}}$

$$
\operatorname{Regret}(K; \pi^\star) \leq \frac{\tau \log|\mathcal{A}|}{\eta} + \frac{\tau + 1}{1-\gamma} + \frac{\eta K}{(1-\gamma)^2} + 12\beta K \sqrt{\frac{d}{\tau}} \log\left(\frac{2K}{\tau\delta}\right) + \frac{\sqrt{2\eta} K}{(1-\gamma)^2 \tau}
$$

$\square$

### D.3 Proof of Theorem 5

To improve the readability of the proof we restate Algorithm 3 hereafter.

---

**Algorithm 5 ILARL** (More readable version).

---

1: **Input:** Dataset size $\tau$, Exploration parameter $\beta$, Step size $\eta$, Expert dataset $\mathcal{D}_{\pi_\mathrm{E}} = \{\boldsymbol{\tau}_i\}_{i=1}^{\tau_E}$.
2: Estimate for expert feature visitation $\widehat{\boldsymbol{\Phi}^\intercal d^{\pi_\mathrm{E}}} \triangleq \frac{(1-\gamma)}{\tau_E} \sum_{\boldsymbol{\tau} \in \mathcal{D}_{\pi_\mathrm{E}}} \sum_{s_h, a_h \in \boldsymbol{\tau}} \gamma^h \phi(s_h, a_h)$ .
3: Initialize $\pi_0$ as uniform distribution over $\mathcal{A}$
4: Initialize $V^1 = 0$
5: **for** $j = 1, \ldots \lfloor K/\tau \rfloor$ **do**
6:     // Collect on-policy data
7:     Denote the indices interval $T_j \triangleq [(j-1) \lfloor K/\tau \rfloor, j \lfloor K/\tau \rfloor)$.
8:     Sample $\mathcal{D}^{(j)} = \{s^i, a^i, s'^{,i}\}_{i \in T_j} \sim d^{\pi^{(j)}}$
9:     Compute $\Lambda^{(j)} = \sum_{(s,a) \in \mathcal{D}^{(j)}} \phi(s,a)\phi(s,a)^\intercal + I$ .
10:     Compute $b^{(j)}(s,a) = \beta \|\phi(s,a)\|_{(\Lambda^{(j)})^{-1}}$.
11:     **for** $k \in T_j$ **do**
12:         // Cost update
13:         Estimate features expectation vector $\widehat{\boldsymbol{\Phi}^\intercal d^{\pi^k}}$ as $\tau^{-1} \sum_{s,a \in \mathcal{D}^{(j)}} \phi(s,a)$.
14:         $\mathbf{w}^{k+1} = \Pi_{\mathcal{W}} \left[ \mathbf{w}^k - \alpha(\widehat{\boldsymbol{\Phi}^\intercal d^{\pi_\mathrm{E}}} - \widehat{\boldsymbol{\Phi}^\intercal d^{\pi^k}}) \right]$ with $\mathcal{W} = \{\mathbf{w} : \|\mathbf{w}\|_2 \leq 1\}$.
15:         // Optimistic Policy Evaluation
16:         $\mathbf{v}^k = (\Lambda^{(j)})^{-1} \sum_{(s,a,s') \in \mathcal{D}^{(j)}} \phi(s,a)V^k(s')$
17:         $Q^{k+1} = \left[ \boldsymbol{\Phi}\mathbf{w}^k + \gamma\boldsymbol{\Phi}\mathbf{v}^k - b^{(j)} \right]_0^{(1-\gamma)^{-1}}$
18:         $V^{k+1}(s) = \langle \pi^{(j)}(a|s), Q^{k+1}(s,a) \rangle$ (notice that $\pi^{(j)} = \pi^{k+1}$).
19:     **end for**
20:     // Policy Improvement Step
21:     Compute average $Q$ value $\bar{Q}^{(j)}(s,a) = \frac{1}{\tau} \sum_{k \in T_j} Q^k(s,a)$.
22:     Update policy

$$\pi^{(j+1)}(a|s) \propto \exp\left( -\eta \sum_{i=1}^{j} \bar{Q}^{(i)}(s,a) \right)$$

23: **end for**

---

*Proof.* Consider the following decomposition

$$\sum_{k=1}^{K} \left\langle \mathbf{c}_{\mathrm{true}}, d^{\pi_\mathrm{E}} - d^{\pi^k} \right\rangle = \sum_{k=1}^{K} \left\langle w_{\mathrm{true}} - w^k, \widehat{\boldsymbol{\Phi}^\intercal d^{\pi_\mathrm{E}}} - \boldsymbol{\Phi}^\intercal d^{\pi^k} \right\rangle + \sum_{k=1}^{K} \left\langle c^k, d^{\pi_\mathrm{E}} - d^{\pi^k} \right\rangle \quad (6)$$

$$+ \sum_{k=1}^{K} \left\langle w_{\mathrm{true}} - w^k, \widehat{\boldsymbol{\Phi}^\intercal d^{\pi_\mathrm{E}}} - \boldsymbol{\Phi}^\intercal d^{\pi_\mathrm{E}} \right\rangle \quad (7)$$

For the first term we can use the following steps.

$$\sum_{k=1}^{K} \left\langle w_{\mathrm{true}} - w^k, \widehat{\boldsymbol{\Phi}^\intercal d^{\pi_\mathrm{E}}} - \boldsymbol{\Phi}^\intercal d^{\pi^k} \right\rangle \leq \sum_{k=1}^{K} \left\langle w_{\mathrm{true}} - w^k, \widehat{\boldsymbol{\Phi}^\intercal d^{\pi_\mathrm{E}}} - \widehat{\boldsymbol{\Phi}^\intercal d^{\pi^k}} \right\rangle + \sum_{k=1}^{K} \left\langle w_{\mathrm{true}} - w^k, \widehat{\boldsymbol{\Phi}^\intercal d^{\pi^k}} - \boldsymbol{\Phi}^\intercal d^{\pi^k} \right\rangle$$

Now, using the regret bound for OMD (Orabona, 2023, Theorem 6.10) we can bound the first term in the decomposition above as

$$\sum_{k=1}^{K} \left\langle w_{\mathrm{true}} - w^k, \widehat{\boldsymbol{\Phi}^\intercal d^{\pi_\mathrm{E}}} - \widehat{\boldsymbol{\Phi}^\intercal d^{\pi^k}} \right\rangle \leq \frac{\max_{w \in \mathcal{W}} \|w - w^1\|_2^2}{2\alpha} + \frac{\alpha}{2} \sum_{k=1}^{K} \left\| \widehat{\boldsymbol{\Phi}^\intercal d^{\pi_\mathrm{E}}} - \widehat{\boldsymbol{\Phi}^\intercal d^{\pi^k}} \right\|_2^2$$

$$\leq \frac{\max_{c \in \mathcal{C}} \|c - c^1\|_2^2}{2\alpha} + \frac{\alpha}{2} \sum_{k=1}^{K} \left\| \widehat{\boldsymbol{\Phi}^\intercal d^{\pi_\mathrm{E}}} - \widehat{\boldsymbol{\Phi}^\intercal d^{\pi^k}} \right\|_1^2$$

$$\leq \frac{1}{2\alpha} + 2\alpha K$$

Then, for $\alpha = \frac{1}{2\sqrt{K}}$, then $\sum_{k=1}^{K} \left\langle w_{\text{true}} - w^k, \widehat{\boldsymbol{\Phi}^\intercal d^{\pi_{\text{E}}}} - \widehat{\boldsymbol{\Phi}^\intercal d^{\pi^k}} \right\rangle \leq 2\sqrt{K}$. For the estimation term,

$$\sum_{k=1}^{K} \left\langle w_{\text{true}} - w^k, \boldsymbol{\Phi}^\intercal d^{\pi^k} - \widehat{\boldsymbol{\Phi}^\intercal d^{\pi^k}} \right\rangle \leq \sum_{k=1}^{K} \left\| w - w^k \right\|_1 \left\| \boldsymbol{\Phi}^\intercal d^{\pi^k} - \widehat{\boldsymbol{\Phi}^\intercal d^{\pi^k}} \right\|_\infty$$

$$\leq \sqrt{d} \sum_{k=1}^{K} \left\| w - w^k \right\|_2 \left\| \boldsymbol{\Phi}^\intercal d^{\pi^k} - \widehat{\boldsymbol{\Phi}^\intercal d^{\pi^k}} \right\|_\infty$$

$$\leq 2\sqrt{d} \sum_{k=1}^{K} \left\| \boldsymbol{\Phi}^\intercal d^{\pi^k} - \widehat{\boldsymbol{\Phi}^\intercal d^{\pi^k}} \right\|_\infty$$

$$\leq 2\sqrt{d} \sum_{k=1}^{K} \sqrt{\frac{2 \log(2dK/\delta)}{\tau}} = 2K \sqrt{\frac{2d \log(2dK/\delta)}{\tau}},$$

where the last inequality holds with probability $1 - \delta$ thanks to Azuma-Hoeffding inequality. Therefore, using Theorem 4 to control the second term in Equation (7) and a union bound we obtain that with probability $1 - 3\delta$.

$$\sum_{k=1}^{K} \left\langle \mathbf{c}_{\text{true}}, d^{\pi_{\text{E}}} - d^{\pi^k} \right\rangle \leq 2\sqrt{K} + 2K \sqrt{\frac{2d \log(2dK/\delta)}{\tau}} + \frac{\tau \log |\mathcal{A}|}{\eta} + \frac{\tau + 1}{1 - \gamma} + \frac{\eta K}{(1 - \gamma)^2}$$

$$+ 12\beta K \sqrt{\frac{d}{\tau}} \log\left(\frac{2K}{\tau\delta}\right) + \frac{\sqrt{2\eta} K}{(1 - \gamma)^2 \tau} + + \sum_{k=1}^{K} \left\langle w_{\text{true}} - w^k, \widehat{\boldsymbol{\Phi}^\intercal d^{\pi_{\text{E}}}} - \boldsymbol{\Phi}^\intercal d^{\pi_{\text{E}}} \right\rangle$$

and using that for the empirical expert an application of Lemma 7 gives that with probability $1 - \delta$.

$$\sum_{k=1}^{K} \left\langle w_{\text{true}} - w^k, \widehat{\boldsymbol{\Phi}^\intercal d^{\pi_{\text{E}}}} - \boldsymbol{\Phi}^\intercal d^{\pi_{\text{E}}} \right\rangle \leq 2K\sqrt{d} \left\| \boldsymbol{\Phi}^\intercal d^{\pi_{\text{E}}} - \widehat{\boldsymbol{\Phi}^\intercal d^{\pi_{\text{E}}}} \right\|_\infty \leq 2K \sqrt{\frac{2d \log(d/\delta)}{\tau_E}}$$

Therefore, selecting $\tau_E \geq \frac{8d \log(d/\delta)}{\epsilon_E^2}$ and using a last union bound gives that with probability $1 - 4\delta$

$$\left\langle \mathbf{c}_{\text{true}}, d^{\pi_{\text{E}}} - \frac{1}{K} \sum_{k=1}^{K} d^{\pi^k} \right\rangle \leq \epsilon_E + \frac{2}{\sqrt{K}} + \sqrt{\frac{8d \log(2dK/\delta)}{\tau}} + \frac{\tau \log |\mathcal{A}|}{\eta K} + \frac{\tau + 1}{(1 - \gamma)K} + \frac{\eta}{(1 - \gamma)^2}$$

$$+ 12\beta \sqrt{\frac{d}{\tau}} \log\left(\frac{2K}{\tau\delta}\right) + \frac{\sqrt{2\eta}}{(1 - \gamma)^2 \tau}$$

Using $\eta = \sqrt{\frac{\tau \log |\mathcal{A}| (1 - \gamma)^2}{K}}$,

$$\left\langle \mathbf{c}_{\text{true}}, d^{\pi_{\text{E}}} - \frac{1}{K} \sum_{k=1}^{K} d^{\pi^k} \right\rangle \leq \epsilon_E + \frac{2}{\sqrt{K}} + \sqrt{\frac{8d \log(2dK/\delta)}{\tau}} + \frac{2}{(1 - \gamma)} \sqrt{\frac{\tau \log |\mathcal{A}|}{K}} + \frac{\tau + 1}{(1 - \gamma)K}$$

$$+ 12\beta \sqrt{\frac{d}{\tau}} \log\left(\frac{2K}{\tau\delta}\right) + \frac{\sqrt{2}}{(1 - \gamma)^2 \tau} \sqrt[4]{\frac{\tau \log |\mathcal{A}| (1 - \gamma)^2}{K}}$$

Neglecting lower order terms we obtain

$$\left\langle \mathbf{c}_{\text{true}}, d^{\pi_{\text{E}}} - \frac{1}{K} \sum_{k=1}^{K} d^{\pi^k} \right\rangle \leq \epsilon_E + \mathcal{O}\left( \sqrt{\frac{8d \log(2dK/\delta)}{\tau}} + \frac{2}{(1 - \gamma)} \sqrt{\frac{\tau \log |\mathcal{A}|}{K}} + 12\beta \sqrt{\frac{d}{\tau}} \log\left(\frac{2K}{\tau\delta}\right) \right)$$

$$\leq \epsilon_E + \mathcal{O}\left( \frac{2}{(1 - \gamma)} \sqrt{\frac{\tau \log |\mathcal{A}|}{K}} + 15\beta \sqrt{\frac{d}{\tau}} \log\left(\frac{2dK}{\tau\delta}\right) \right)$$

Therefore, choosing $\tau = \mathcal{O}\left( \frac{\beta(1 - \gamma)\sqrt{dK} \log(2dK/\delta)}{\sqrt{\log |\mathcal{A}|}} \right)$, gives

$$\left\langle \mathbf{c}_{\text{true}}, d^{\pi_{\text{E}}} - \frac{1}{K} \sum_{k=1}^{K} d^{\pi^k} \right\rangle \leq \epsilon_E + \widetilde{\mathcal{O}}\left( \frac{\log^{1/4} |\mathcal{A}| \, d^{1/4} \sqrt{\beta}}{\sqrt{1 - \gamma} K^{1/4}} \right)$$

Therefore, choosing $K \geq \widetilde{\mathcal{O}}\left(\frac{\log|\mathcal{A}|d\beta^2 \log^2(2dK/\delta)}{(1-\gamma)^2\epsilon^4}\right)$ which is attained by $K = \mathcal{O}\left(\frac{\log|\mathcal{A}|d\beta^2}{(1-\gamma)^2\epsilon^4}\log^2(\frac{d\beta\log|\mathcal{A}|}{\delta(1-\gamma)\epsilon})\right)$ ensures $\left\langle \mathbf{c}_{\text{true}}, d^{\pi_{\text{E}}} - \frac{1}{K}\sum_{k=1}^{K} d^{\pi^k}\right\rangle \leq \epsilon + \epsilon_E$ with probability $1 - 4\delta$. $\qquad\square$

## E  TECHNICAL LEMMAS

**Lemma 2.** *Assume that in Algorithm 1, we set $\beta = \mathcal{O}\left(dH\log(RdH\delta^{-1})\right)$ with $R = \tau^2\sqrt{d}H$. Then, with probability $1 - \delta$ it holds that*

$$-2b_h^k(s,a) \leq Q_h^k(s,a) - \mathbf{c}_h^k(s,a) - PV_{h+1}^k(s,a) \leq 0 \quad \forall s,a \in \mathcal{S} \times \mathcal{A}, k \in [K], h \in [H]$$

*Proof.* We first see that Lemma 4 holds for every $j$. Then, thanks to union bound we have that with probability $1 - \delta$ it holds that for any state action pairs we have that

$$\left|\phi(s,a)^T\mathbf{v}_h^k - PV_{h+1}^k(s,a)\right| \leq \beta\left\|\phi(s,a)\right\|_{(\Lambda_h^{(j)})^{-1}} = b_h^k(s,a)$$

From this fact the conclusion follows immediately if no truncation happens. That is , if $Q_h^k = \mathbf{c}_h^k + \mathbf{\Phi}\mathbf{v}_h^k - b_h^k$. Now, we consider the case where a lower truncation takes place, in this case, we have

$$Q_h^k = 0 \leq \mathbf{c}_h^k + PV_{h+1}^k$$

If there a truncation from above it holds that

$$Q_h^k \leq \mathbf{c}_h^k + \mathbf{\Phi}\mathbf{v}_h^k - b_h^k \leq \mathbf{c}_h^k + PV_{h+1}^k + b_h^k - b_h^k = \mathbf{c}_h^k + PV_{h+1}^k$$

To show the lower bound in the lemma in case of a lower truncation, we have that

$$Q_h^k \geq \mathbf{c}_h^k + \mathbf{\Phi}\mathbf{v}_h^k - b_h^k \geq \mathbf{c}_h^k + PV_{h+1}^k - b_h^k - b_h^k = \mathbf{c}_h^k + PV_{h+1}^k - 2b_h^k$$

finally, if the truncation from above is triggered, we have that

$$Q_h^k = H - h + 1 \geq \mathbf{c}_h^k + PV_{h+1}^k \geq \mathbf{c}_h^k + PV_{h+1}^k - 2b_h^k$$

$\qquad\square$

**Lemma 3.** *For any $k \in [K]$, let the bonus $b^k$ be defined as in Algorithm 2 with $\beta = \mathcal{O}\left(d(1-\gamma)^{-1}\log(Rd(1-\gamma)^{-1}\delta^{-1})\right)$ with $R = \tau^2\sqrt{d}(1-\gamma)^{-1}$. Then, with probability $1 - \delta$ it holds that*

$$-2b^k(s,a) \leq Q^{k+1}(s,a) - \mathbf{c}^k(s,a) - \gamma PV^k(s,a) \leq 0 \quad \forall s,a \in \mathcal{S} \times \mathcal{A}, k \in [K]$$

*Proof.* We first see that Lemma 5 holds for every $j$. Then, thanks to union bound we have that with probability $1 - \delta$ it holds that for any state action pairs we have that

$$\left|\phi(s,a)^T\mathbf{v}^k - PV^k(s,a)\right| \leq \beta\left\|\phi(s,a)\right\|_{(\Lambda^{(j)})^{-1}} = b^k(s,a)$$

From this fact the conclusion follows immediately if no truncation happens. That is , if $Q^{k+1} = \mathbf{c}^k + \mathbf{\Phi}\mathbf{v}^k - b^k$. Now, we consider the case where a upper truncation takes place, in this case, we have

$$Q^{k+1} = \frac{1}{1-\gamma} = 1 + \frac{\gamma}{1-\gamma} \geq \mathbf{c}^k + \gamma PV^k - 2b^k$$

While for the upper bound, we have that

$$Q^{k+1} \leq \mathbf{c}^k + \gamma\mathbf{\Phi}\mathbf{v}^k - b^k \leq \mathbf{c}^k + \gamma PV^k + b^k - b^k = \mathbf{c}^k + \gamma PV^k$$

Now, we handle the case of a lower truncation in this case

$$Q^{k+1} \geq \mathbf{c}^k + \gamma\mathbf{\Phi}\mathbf{v}^k - b^k \geq \mathbf{c}^k + \gamma PV^k - b^k - b^k = \mathbf{c}^k + \gamma PV^k - 2b^k$$

for the upper bound we have that

$$Q^{k+1} = 0 \leq \mathbf{c}^k + \gamma PV^k$$

$\qquad\square$

**Lemma 4.** *Let $V_h^k$ be the sequence of value functions generated by Algorithm 1, fix a batch index $j$ and let $T_j$ denote the set of indices in the $j^{th}$ batch. Then, it holds that for $\beta = \widetilde{\mathcal{O}}(dH)$, the estimator*

$$\mathbf{v}_h^k = (\Lambda^{(j)})^{-1} \sum_{(s,a,s') \in \mathcal{D}_h^{(j)}} \phi(s,a) V_{h+1}^k(s')$$

*satisfies*

$$\left| \phi(s,a)^T \mathbf{v}_h^k - PV_{h+1}^k(s,a) \right| \leq \beta \left\| \phi(s,a) \right\|_{(\Lambda_h^{(j)})^{-1}} = b^k(s,a) \quad \forall k \in T_j, h \in [H], \forall s,a \in \mathcal{S} \times \mathcal{A}$$

*with probability $1 - \delta\tau/K$.*

*Proof.* The proof is analogous to the proof of Lemma 5 but invoking Theorem 7 with $B = H$ and applying a further union bound over the set $[H]$. Thus, the proof is skipped for brevity. $\qquad\square$

**Lemma 5.** *Let $V^k$ be the sequence of value functions generated by Algorithm 2, fix a batch index $j$ and let $T_j$ denote the set of indices in the $j^{th}$ batch. Then, it holds that for $\beta = \widetilde{\mathcal{O}}\left(\frac{d}{1-\gamma}\right)$, the estimator*

$$\mathbf{v}^k = (\Lambda^{(j)})^{-1} \sum_{(s,a,s') \in \mathcal{D}^{(j)}} \phi(s,a) V^k(s')$$

*satisfies*

$$\left| \phi(s,a)^T \mathbf{v}^k - PV^k(s,a) \right| \leq \beta \left\| \phi(s,a) \right\|_{(\Lambda^{(j)})^{-1}} = b^k(s,a) \quad \forall k \in T_j, \forall s,a \in \mathcal{S} \times \mathcal{A}$$

*with probability $1 - \delta\tau/K$.*

*Proof.* With standard manipulation one can prove that

$$PV^k = \mathbf{\Phi}MV^k = \mathbf{\Phi}(\Lambda^{(j)})^{-1}MV^k + \mathbf{\Phi}(\Lambda^{(j)})^{-1} \sum_{s,a,s' \in \mathcal{D}^{(j)}} \phi(s,a)PV^k(s,a)$$

and by definition

$$\mathbf{\Phi}\mathbf{v}^k = \mathbf{\Phi}(\Lambda^{(j)})^{-1} \sum_{s,a,s' \in \mathcal{D}^{(j)}} \phi(s,a)V^k(s')$$

Therefore,

$$PV^k - \mathbf{\Phi}\mathbf{v}^k = \mathbf{\Phi}(\Lambda^{(j)})^{-1}MV^k + \mathbf{\Phi}(\Lambda^{(j)})^{-1} \sum_{s,a,s' \in \mathcal{D}^{(j)}} \phi(s,a)(PV^k(s,a) - V^k(s'))$$

Then, for any state action pair $(s,a)$, we have

$$\left| PV^k(s,a) - \mathbf{\Phi}\mathbf{v}^k(s,a) \right| = \phi(s,a)^T(\Lambda^{(j)})^{-1}MV^k + \phi(s,a)^T(\Lambda^{(j)})^{-1} \sum_{s,a,s' \in \mathcal{D}^{(j)}} \phi(s,a)(PV^k(s,a) - V^k(s'))$$

by applying Holder's inequality,

$$\left| PV^k(s,a) - \mathbf{\Phi}\mathbf{v}^k(s,a) \right| \leq \left\| \phi(s,a) \right\|_{(\Lambda^{(j)})^{-1}} \left\| (\Lambda^{(j)})^{-1}MV^k \right\|_{(\Lambda^{(j)})}$$

$$+ \left\| \phi(s,a) \right\|_{(\Lambda^{(j)})^{-1}} \left\| (\Lambda^{(j)})^{-1} \sum_{s,a,s' \in \mathcal{D}^{(j)}} \phi(s,a)(PV^k(s,a) - V^k(s')) \right\|_{(\Lambda^{(j)})}$$

$$= \left\| \phi(s,a) \right\|_{(\Lambda^{(j)})^{-1}} \left\| MV^k \right\|_{(\Lambda^{(j)})^{-1}}$$

$$+ \left\| \phi(s,a) \right\|_{(\Lambda^{(j)})^{-1}} \left\| \sum_{s,a,s' \in \mathcal{D}^{(j)}} \phi(s,a)(PV^k(s,a) - V^k(s')) \right\|_{(\Lambda^{(j)})^{-1}}$$

where in the equality we used that for a symmetric matrix $A$ we have that $\|Ax\|_{A^{-1}} = \|x\|_A$. Then, we can use that $\|MV^k\|_{(\Lambda^{(j)})^{-1}} \leq \frac{1}{1-\gamma}$ to obtain

$$\left|PV^k(s,a) - \boldsymbol{\Phi}\mathbf{v}^k(s,a)\right| \leq \|\phi(s,a)\|_{(\Lambda^{(j)})^{-1}} \left( \frac{1}{1-\gamma} + \left\| \sum_{s,a,s' \in \mathcal{D}^{(j)}} \phi(s,a)(PV^k(s,a) - V^k(s')) \right\|_{(\Lambda^{(j)})^{-1}} \right)$$

To handle the second term in brackets we use that $V^k \in \mathcal{V}^{\pi^{(j)}}$ defined as

$$\mathcal{V}^{\pi^{(j)}} = \left\{ \left\langle \pi^{(j)}(\cdot|s), Q(s,\cdot) \right\rangle | Q(s,a) \in \mathcal{Q}(\beta, \Lambda, \mathbf{w}, \mathbf{v}) \right\}$$

where $\mathcal{Q}(\beta, \Lambda, \mathbf{w}, \mathbf{v})$ is defined as in Theorem 8. Denote as $\mathcal{N}_\epsilon^j$ the $\|\cdot\|_\infty$-covering number of the class $\mathcal{V}^{\pi^{(j)}}$ and notice that $\mathcal{V}^{\pi^{(j)}}$ and $\mathcal{D}^{(j)}$ are conditionally independent given $\pi^{(j)}$.

Under this setting we can use Theorem 7 with $B = (1-\gamma)^{-1}$ to obtain that with probability $1-\delta\tau/K$

$$\left|PV^k(s,a) - \boldsymbol{\Phi}\mathbf{v}^k(s,a)\right| \leq \|\phi(s,a)\|_{(\Lambda^{(j)})^{-1}} \left( \frac{1}{1-\gamma} + \sqrt{\frac{2d}{(1-\gamma)^2} \log\left(\frac{K(\tau+1)}{\delta\tau}\right) + \frac{4}{(1-\gamma)^2} \log \mathcal{N}_\epsilon^j + 8\tau^2\epsilon^2} \right)$$

$$\leq \frac{\|\phi(s,a)\|_{(\Lambda^{(j)})^{-1}}}{1-\gamma} \left( 1 + \sqrt{2d \log\left(\frac{K(\tau+1)}{\delta\tau}\right)} + 2\sqrt{\log \mathcal{N}_\epsilon^j} + 2\sqrt{2}\tau\epsilon \right).$$

Then, we can conclude that the covering number is upper bounded by Theorem 8 with $L = \frac{\tau}{1-\gamma}$ since

$$\|\mathbf{v}^k\| \leq \frac{1}{(1-\gamma)} \left\| (\Lambda^{(j)})^{(-1)} \right\| \left\| \sum_{s,a,s' \in \mathcal{D}^{(j)}} \phi(s,a) \right\| \leq \frac{\tau}{(1-\gamma)}$$

we obtain that

$$\log \mathcal{N}_\epsilon^j \leq d \log\left( 1 + \frac{4}{\epsilon}\sqrt{1 + \frac{\gamma^2\tau^2}{(1-\gamma)^2}} \right) + d^2 \log(1 + 8\sqrt{d}\beta^2\epsilon^{-2})$$

Therefore,

$$\left|PV^k(s,a) - \boldsymbol{\Phi}\mathbf{v}^k(s,a)\right| \leq \frac{\|\phi(s,a)\|_{(\Lambda^{(j)})^{-1}}}{1-\gamma} \left[ 1 + \sqrt{2d \log\left(\frac{K(\tau+1)}{\delta\tau}\right)} + \sqrt{2d \log\left(1 + \frac{4}{\epsilon}\sqrt{1 + \frac{\gamma^2\tau^2}{(1-\gamma)^2}}\right)} \right.$$

$$\left. + \sqrt{2}d\sqrt{\log(1 + 8\sqrt{d}\beta^2\epsilon^{-2})} + 2\sqrt{2}\tau\epsilon \right]$$

At this point, using $\epsilon = \tau^{-1}$, we obtain

$$\left|PV^k(s,a) - \boldsymbol{\Phi}\mathbf{v}^k(s,a)\right| \leq \frac{\|\phi(s,a)\|_{(\Lambda^{(j)})^{-1}}}{1-\gamma} \left[ 1 + \sqrt{2d \log\left(\frac{K(\tau+1)}{\delta\tau}\right)} + \sqrt{2d \log\left(1 + 4\tau\sqrt{1 + \frac{\gamma^2\tau^2}{(1-\gamma)^2}}\right)} \right.$$

$$\left. + \sqrt{2}d\sqrt{\log(1 + 8\sqrt{d}\beta^2\tau^2)} + 2\sqrt{2} \right]$$

To simplify the above expression, we notice that there exists a constant $c$ such that

$$\left|PV^k(s,a) - \boldsymbol{\Phi}\mathbf{v}^k(s,a)\right| \leq c \frac{\|\phi(s,a)\|_{(\Lambda^{(j)})^{-1}}}{1-\gamma} d\sqrt{\log\left(\frac{\tau K\beta\sqrt{d}}{\delta(1-\gamma)}\right)}$$

Now, using (Sherman et al., 2023b, Lemma D.2), we have that $\beta = \mathcal{O}(d(1-\gamma)^{-1} \log\left[Rd(1-\gamma)^{-1}\right])$ with $R = \tau K\sqrt{d}\delta^{-1}(1-\gamma)^{-1}$ ensures

$$\beta \geq c\frac{d}{(1-\gamma)} \log\left(\frac{\tau^2\beta\sqrt{d}}{\delta(1-\gamma)}\right),$$

and therefore

$$\left| PV^k(s,a) - \mathbf{\Phi}\mathbf{v}^k(s,a) \right| \leq \beta \left\| \phi(s,a) \right\|_{(\Lambda^{(j)})^{-1}}.$$

□

**Theorem 7.** *For a fixed policy $\pi$ consider a function class $\mathcal{V}$ and a state action pair dataset $\mathcal{D}$ collected with a fixed policy $\pi$ such that $\mathcal{D}$ and $\mathcal{V}$ are conditionally independent given $\pi$. Then, for any $f \in \mathcal{V}$ such that $\|f\|_\infty \leq B$, it holds with probability $1 - \delta\tau/K$*

$$\left\| \sum_{s,a,s' \in \mathcal{D}} \phi(s,a)(Pf(s,a) - f(s')) \right\|_{(\Lambda^{(j)})^{-1}}^2 \leq 2dB^2 \log\left( \frac{K(\tau+1)}{\delta\tau} \right) + 4B^2 \log \mathcal{N}_\epsilon + 8\tau^2\epsilon^2$$

*where $N_\epsilon$ is the the $(\epsilon, \|\cdot\|_\infty)$- covering number of the class $\mathcal{V}^\pi$.*

*Proof.* Consider the decomposition in (Jin et al., 2019, Lemma D.4). In particular, let $\mathcal{C}_\epsilon(\mathcal{V})$ denote the $(\epsilon, \|\cdot\|_\infty)$-covering set of $\mathcal{V}$ and pick $\tilde{f} \in \mathcal{C}_\epsilon(\mathcal{V})$ such that $\left\| f - \tilde{f} \right\|_\infty \leq \epsilon$. The existence of $\tilde{f}$ is guaranteed by the properties of covering sets. Then, we have

$$\left\| \sum_{s,a,s' \in \mathcal{D}} \phi(s,a)(Pf(s,a) - f(s')) \right\|_{(\Lambda^{(j)})^{-1}}^2 \leq 2\left\| \sum_{s,a,s' \in \mathcal{D}} \phi(s,a)(P\tilde{f}(s,a) - \tilde{f}(s')) \right\|_{(\Lambda^{(j)})^{-1}}^2$$

$$+ 2\left\| \sum_{s,a,s' \in \mathcal{D}} \phi(s,a)(P(f - \tilde{f})(s,a) - (f - \tilde{f})(s')) \right\|_{(\Lambda^{(j)})^{-1}}^2$$

The second term can be bounded by $8\tau^2\epsilon^2$ as in Jin et al. (2019) so now we focus on the first term via a uniform bound over the set $\mathcal{C}_\epsilon(\mathcal{V})$. We need to index the dataset $\mathcal{D}$, i.e. $\mathcal{D} = \left\{ (s^\ell, a^\ell) \right\}_{\ell=1}^{|\mathcal{D}|}$ and consider the filtration $\mathcal{F}_j = \left\{ (s^\ell, a^\ell) \right\}_{\ell=1}^j$. Since the features mapping is deterministic, $\phi(s^\ell, a^\ell)$ is $\mathcal{F}_\ell$-measurable. Then, notice that by assumption $\mathcal{D}$ and $\mathcal{V}^\pi$ are conditionally independent given $\pi$. Therefore, we also have that $\mathcal{D}$ and $\mathcal{C}_\epsilon(\mathcal{V}^\pi)$ are conditionally independent given $\pi$. So for any function $\bar{f} \in \mathcal{C}_\epsilon(\mathcal{V}^\pi)$ we have that $\mathbb{E}[\bar{f}(s^{\ell+1})|\mathcal{F}_\ell] = P\bar{f}(s^\ell, a^\ell)$. Finally, from the assumption $\left\| \bar{f} \right\|_\infty \leq B$ we have that $\bar{f}$ is $B^2$-subgaussian. Therefore, all the conditions of (Jin et al., 2019, Theorem D.3) are met and via a union bound over the covering set allows to conclude that with probability $1 - \delta\tau/K$

$$2\left\| \sum_{s,a,s' \in \mathcal{D}} \phi(s,a)(P\bar{f}(s,a) - \bar{f}(s')) \right\|_{(\Lambda^{(j)})^{-1}}^2 \leq 2dB^2 \log\left( \frac{K(\tau+1)}{\delta\tau} \right) + 4B^2 \log \mathcal{N}_\epsilon \quad \forall \bar{f} \in \mathcal{C}_\epsilon(\mathcal{V}),$$

and since $\tilde{f} \in \mathcal{C}_\epsilon(\mathcal{V})$,

$$2\left\| \sum_{s,a,s' \in \mathcal{D}} \phi(s,a)(P\tilde{f}(s,a) - \tilde{f}(s')) \right\|_{(\Lambda^{(j)})^{-1}}^2 \leq 2dB^2 \log\left( \frac{K(\tau+1)}{\delta\tau} \right) + 4B^2 \log \mathcal{N}_\epsilon.$$

□

**Theorem 8.** *Let us consider the function class $\mathcal{Q}$ defined as follows*

$$\mathcal{Q}(\beta, \Lambda, \mathbf{w}, \mathbf{v}) = \{Q(s,a; \beta, \Lambda, \mathbf{w}, \mathbf{v}) | \beta \in \mathbb{R}, \lambda_{\min}(\Lambda) \geq 1, \|\mathbf{w}\| \leq 1, \|\mathbf{v}\| \leq L\}$$

$$\text{where} \quad Q(s,a; \beta, \Lambda, \mathbf{w}, \mathbf{v}) = [\phi(s,a)^\mathsf{T}(\mathbf{w} + \gamma\mathbf{v}) + \beta \|\phi(s,a)\|_{\Lambda^{-1}}]_0^{(1-\gamma)^{-1}}$$

*and the classes*

$$\mathcal{V}^\pi = \{\langle \pi(\cdot|s), Q(s,\cdot) \rangle | Q(s,a) \in \mathcal{Q}(\beta, \Lambda, \mathbf{w}, \mathbf{v})\}$$

*for any $\pi : \mathcal{S} \to \Delta_\mathcal{A}$.*

*Then, it holds that for any $\pi : \mathcal{S} \to \Delta_\mathcal{A}$*

$$\mathcal{N}_\epsilon(\mathcal{V}^\pi) \leq \mathcal{N}_\epsilon(\mathcal{Q}) = (1 + 4\sqrt{1 + \gamma^2 L^2}/\epsilon)^d (1 + 8\sqrt{d}\beta^2\epsilon^{-2})^{d^2}$$

*Proof.* Let us remove clipping that can only decreasing the covering number of the function class and let us consider the matrix $A = \beta^2 \Lambda^{-1}$, then we can rewrite the function class of interest as parameterized only by $A$ rather then $\beta$ and $\Lambda$ separately. In addition, let us consider a vector $\mathbf{z} = \mathbf{w} + \gamma \mathbf{v}$

$$\mathcal{Q}(A, \mathbf{z}) = \left\{ Q(s, a; A, \mathbf{w}, \mathbf{v}) | \lambda_{\min}(\Lambda) \geq \beta^2, \|\mathbf{z}\|^2 \leq 2 + 2\gamma^2 L^2 \right\}$$

with

$$Q(s, a; A, \mathbf{z}) = \phi(s, a)^{\mathsf{T}} \mathbf{z} + \|\phi(s, a)\|_A$$

Then, we have that

$$|Q(s, a; A_1, \mathbf{z}_1) - Q(s, a; A_2, \mathbf{z}_2)| \leq \|\phi(s, a)\| \|\mathbf{z}_1 - \mathbf{z}_2\| + \left| \sqrt{\phi(s, a)^{\mathsf{T}} A_1 \phi(s, a)} - \sqrt{\phi(s, a)^{\mathsf{T}} A_2 \phi(s, a)} \right|$$

$$\leq \|\phi(s, a)\| \|\mathbf{z}_1 - \mathbf{z}_2\| + \sqrt{|\phi(s, a)^{\mathsf{T}} (A_1 - A_2) \phi(s, a)|}$$

$$\leq \|\mathbf{z}_1 - \mathbf{z}_2\| + \sqrt{\sup_{\phi : \|\phi\| \leq 1} |\phi(s, a)^{\mathsf{T}} (A_1 - A_2) \phi(s, a)|}$$

$$\leq \|\mathbf{z}_1 - \mathbf{z}_2\| + \sqrt{\|A_1 - A_2\|}$$

$$\leq \|\mathbf{z}_1 - \mathbf{z}_2\| + \sqrt{\|A_1 - A_2\|_F}$$

where $\|A_1 - A_2\|$ is the spectral norm of the matrix $A_1 - A_2$ and $\|A_1 - A_2\|_F$ is the Frobenius norm. We also used the inequality $\left| \sqrt{x} - \sqrt{y} \right| \leq \sqrt{|x - y|}$ that holds for any $x, y \geq 0$. At this point we can constructing an $\epsilon$-covering set for $\mathcal{Q}(A, \mathbf{z})$ as product of the $\epsilon^2/4$ covering set for the set $\mathcal{Y} = \left\{ A \in \mathbb{R}^{d \times d} | \|A\|_F \leq \sqrt{d}\beta^{-2} \right\}$ which has cardinality $\mathcal{N}_\epsilon(\mathcal{Y}) = (1 + 8\sqrt{d}\beta^2 \epsilon^{-2})^{d^2}$ while the covering for the set $\mathcal{Z} = \left\{ \mathbf{z} \in \mathbb{R}^d : \|\mathbf{z}\|^2 \leq 1 + \gamma^2 L^2 \right\}$ satisfies $\mathcal{N}_\epsilon(\mathcal{Z}) = (1 + 4\sqrt{1 + \gamma^2 L^2}/\epsilon)^d$. Hence, taking the product, we have that

$$\mathcal{N}_\epsilon(\mathcal{Q}) = (1 + 4\sqrt{1 + \gamma^2 L^2}/\epsilon)^d (1 + 8\sqrt{d}\beta^2 \epsilon^{-2})^{d^2}$$

.

At this point, let us consider the set

$$\mathcal{V}^\pi = \{ \langle \pi(\cdot|s), Q(s, \cdot) \rangle | Q(s, a) \in \mathcal{Q}(A, \mathbf{z}) \}$$

Since the policy $\pi$ is fixed and averaging is a non expansive operation, we have that $\mathcal{N}_\epsilon(\mathcal{V}^\pi) \leq \mathcal{N}_\epsilon(\mathcal{Q})$.

However, for the set

$$\mathcal{V} = \{ \langle \pi(\cdot|s), Q(s, \cdot) \rangle | \pi \in \Pi, Q(s, a) \in \mathcal{Q}(\beta, \Lambda, \mathbf{w}, \mathbf{v}) \}$$

the averaging is not wrt to a fixed distribution therefore we would need to proceed as follow

$$|\langle \pi_1(\cdot|s), Q_1(s, \cdot) \rangle - \langle \pi_2(\cdot|s), Q_2(s, \cdot) \rangle| \leq \frac{1}{(1 - \gamma)} \|\pi_1(\cdot|s) - \pi_2(\cdot|s)\|_1 + 2 \|Q_1(s, \cdot) - Q_2(s, \cdot)\|_\infty$$

$$\leq \frac{1}{(1 - \gamma)} \|\pi_1(\cdot|s) - \pi_2(\cdot|s)\|_1 + 2 \|Q_1(s, \cdot) - Q_2(s, \cdot)\|_\infty$$

$$\leq \frac{1}{(1 - \gamma)} \max_{s \in \mathcal{S}} \|\pi_1(\cdot|s) - \pi_2(\cdot|s)\|_1 + 2 \|\mathbf{z}_1 - \mathbf{z}_2\| + \sqrt{\|A_1 - A_2\|_F}$$

$$\leq \frac{1}{(1 - \gamma)} \|\pi_1(\cdot|s) - \pi_2(\cdot|s)\|_{\infty, 1} + 2 \|\mathbf{z}_1 - \mathbf{z}_2\| + \sqrt{\|A_1 - A_2\|_F}$$

Therefore, we can conclude that $\mathcal{N}_\epsilon(\mathcal{V}) = \mathcal{N}_\epsilon(\Pi, \|\cdot\|_{\infty, 1}) \mathcal{N}_\epsilon(\mathcal{Q})$. $\square$

Next, we prove the Lemma that we use to state Theorems 1 and 2 using $\delta = \epsilon$.

**Lemma 6.** *High probability to expectation conversion for a bounded random variable Let us consider a random variable $X$ such that $-X_{\max} \leq X \leq X_{\max}$ almost surely and that $\mathbb{P}[X \geq \mu] \leq \delta$, then it holds that*

$$\mathbb{E}[X] \leq \mu + \delta(X_{\max} - \mu)$$

*Proof.*

$$\mathbb{E}\left[X\right] = (1-\delta)\mathbb{E}\left[X|X \leq \mu\right] + \delta\mathbb{E}\left[X|X \geq \mu\right] \leq (1-\delta)\mu + \delta X_{\max}$$

$\square$

**Lemma 7.** ***Expert concentration*** *(Syed & Schapire, 2007, Theorem 1)* *Let* $\mathcal{D}_{\pi_{\mathrm{E}}} \triangleq \{(s_0^\ell, a_0^\ell, s_1^\ell, a_1^\ell, \ldots, s_H^\ell, a_H^\ell)\}_{\ell=1}^{n_{\mathrm{E}}}$ *be a finite set of i.i.d. truncated sample trajectories collected with an expert policy* $\pi_{\mathrm{E}}$. *We consider the empirical expert feature expectation vector* $\mathbf{\Phi}^{\intercal} d^{\pi_{\mathrm{E}}}$ *by taking sample averages, i.e.,*

$$\widehat{\mathbf{\Phi}^{\intercal} d^{\pi_{\mathrm{E}}}} \triangleq (1-\gamma)\frac{1}{n_{\mathrm{E}}}\sum_{t=0}^{H}\sum_{\ell=1}^{N}\gamma^t \phi_i(s_t^\ell, a_t^\ell), \ \forall \ i \in [d].$$

*Suppose the trajectory length is* $H \geq \frac{1}{1-\gamma}\log(\frac{1}{\varepsilon})$, *and the number of of expert trajectories is* $n_{\mathrm{E}} \geq \frac{2\log(\frac{2d}{\delta})}{\varepsilon^2}$. *Then, with probability at least* $1 - \delta$, *it holds that* $\left\|\mathbf{\Phi}^{\intercal} d^{\pi_{\mathrm{E}}} - \widehat{\mathbf{\Phi}^{\intercal} d^{\pi_{\mathrm{E}}}}\right\|_{\infty} \leq \varepsilon$.

# F  OMITTED PROOFS FOR BEST RESPONSE IMITATION LEARNING

## F.1  PROOF OF THEOREM 6

*Proof.*

$$\begin{aligned}
\mathrm{Regret}(K, \pi^\star) &= \sum_{k=1}^{K} V_1^{\pi^k, k}(s_1) - V_1^{\pi^\star, k} \\
&= \sum_{k=1}^{K} V_1^{\pi^k, k}(s_1) - V_1^{k-1}(s_1) + V_1^{k-1}(s_1) - V_1^{\pi^\star, k} \\
&= \sum_{k=1}^{K}\sum_{h=1}^{H} \mathbb{E}_{s,a\sim d_h^{\pi^k}}\left[\mathbf{c}_h^k(s,a) + P_h V_{h+1}^{k-1}(s,a) - Q_h^{k-1}(s,a)\right] \\
&\quad + \sum_{k=1}^{K}\sum_{h=1}^{H} \mathbb{E}_{s,a\sim d_h^{\pi^\star}}\left[Q_h^{k-1}(s,a) - \mathbf{c}_h^k(s,a) - P_h V_{h+1}^{k-1}(s,a)\right] \\
&\quad + \sum_{h=1}^{H} \mathbb{E}_{s\sim d_h^{\pi^\star}}\left[\sum_{k=1}^{K}\left\langle \pi_h^k(\cdot|s) - \pi_h^\star(\cdot|s), Q_h^{k-1}(s,a)\right\rangle\right] \\
&\leq \sum_{k=1}^{K}\sum_{h=1}^{H} \mathbb{E}_{s,a\sim d_h^{\pi^k}}\left[\mathbf{c}_h^k(s,a) + P_h V_{h+1}^{k-1}(s,a) - Q_h^{k-1}(s,a)\right] \\
&\quad + \sum_{k=1}^{K}\sum_{h=1}^{H} \mathbb{E}_{s,a\sim d_h^{\pi^\star}}\left[Q_h^{k-1}(s,a) - \mathbf{c}_h^k(s,a) - P_h V_{h+1}^{k-1}(s,a)\right]
\end{aligned}$$

where the last inequality is due to the use of the best response (greedy policy) in Step 17 of Algorithm 4. At this point we can prove the optimistic properties of the estimator (that follows combining Lemmas 3 and 8),i.e. for any $h = H, \ldots, 1$, it holds that

$$\mathbf{c}_h^k(s,a) + P_h V_{h+1}^{k-1}(s,a) - 2b_h^{k-1}(s,a) \leq Q_h^{k-1}(s,a) \leq \mathbf{c}_h^k(s,a) + P_h V_{h+1}^{k-1}(s,a) \quad \forall s,a \in \mathcal{S}\times\mathcal{A} \quad w.p. \quad 1-\delta.$$

Thus, it holds with probability $1 - \delta$ that

$$\mathrm{Regret}(K, \pi^\star) \leq 2\sum_{k=1}^{K}\sum_{h=1}^{H} \mathbb{E}_{s,a\sim d_h^{\pi^k}}\left[b_h^{k-1}(s,a)\right]$$

and then, using Cauchy-Schwartz and the elliptical potential lemma ( see Lemma 9), we obtain that with probability $1 - 2\delta$,

$$\mathrm{Regret}(K, \pi^\star) \leq \mathcal{O}\left(H^2 d^{3/2}\sqrt{K\log(K\delta^{-1})}\right)$$

Then, we apply this result in the imitation learning setting. We start with our usual decomposition

$$
\sum_{k=1}^{K}\sum_{h=1}^{H}\left\langle \mathbf{c}_{\text{true},h}, d_h^{\pi^k} - d_h^{\widehat{\pi_{\mathrm{E}}}}\right\rangle = \sum_{k=1}^{K}\sum_{h=1}^{H}\left\langle \mathbf{c}_h^k, d_h^{\pi^k} - d_h^{\widehat{\pi_{\mathrm{E}}}}\right\rangle + \sum_{k=1}^{K}\sum_{h=1}^{H}\left\langle \mathbf{c}_{\text{true},h} - \mathbf{c}_h^k, d_h^{\pi^k} - d_h^{\widehat{\pi_{\mathrm{E}}}}\right\rangle
$$

$$
= \sum_{k=1}^{K}\sum_{h=1}^{H}\left\langle \mathbf{c}_h^k, d_h^{\pi^k} - d_h^{\widehat{\pi_{\mathrm{E}}}}\right\rangle + \sum_{k=1}^{K}\sum_{h=1}^{H}\left\langle \mathbf{c}_{\text{true},h} - \mathbf{c}_h^k, \mathbb{1}\left[s_k^h, a_k^h\right] - d_h^{\widehat{\pi_{\mathrm{E}}}}\right\rangle
$$

$$
- \sum_{h=1}^{H}\sum_{k=1}^{K}\left\langle \mathbf{c}_{\text{true},h} - \mathbf{c}_h^k, \mathbb{1}\left[s_k^h, a_k^h\right] - d_h^{\pi^k}\right\rangle
$$

Then, notice that $z_h^k = -\left\langle \mathbf{c}_{\text{true},h} - \mathbf{c}_h^k, \mathbb{1}\left[s_k^h, a_k^h\right] - d_h^{\pi^k}\right\rangle$ is a martingale difference sequence adapted to the filtration $\mathcal{F}_h^k = \left\{\left(\tau^k, \mathbf{c}_h^k\right)\right\}$ almost surely bounded by $4$. Therefore, by Azuma Hoeffding inequality, we obtain $\sum_{h=1}^{H}\sum_{k=1}^{K} z_h^k \leq H\sqrt{8K\log\delta^{-1}}$. Therefore, via a union bound, we have that with probability $1 - 3\delta$, it holds

$$
\sum_{k=1}^{K}\sum_{h=1}^{H}\left\langle \mathbf{c}_{\text{true},h}, d_h^{\pi^k} - d_h^{\widehat{\pi_{\mathrm{E}}}}\right\rangle \leq \widetilde{\mathcal{O}}\left(H^2 d^{3/2}\sqrt{K\log(K\delta^{-1})}\right) + \frac{H}{\alpha} + 2\alpha K H + H\sqrt{8K\log\delta^{-1}}
$$

$$
= \widetilde{\mathcal{O}}\left(H^2 d^{3/2}\sqrt{K\log(K\delta^{-1})}\right) + 4H\sqrt{K} + H\sqrt{8K\log\delta^{-1}}
$$

where last step follows from choosing $\alpha = \frac{1}{\sqrt{2K}}$. At this point, the conclusion holds plugging in the value for $K$ in the statement of the main theorem which is $K = \mathcal{O}\left(\frac{H^4 d^3\log(dH/(\epsilon\delta))}{\epsilon^2}\right)$. Finally, we need to control the error in the estimation of the expert occupancy measure that can be done as in the proof for Algorithm 1.

$$
V_1^{\pi_{\mathrm{E}}}(s_1; \mathbf{c}_{\text{true}}) - V_1^{\widehat{\pi_{\mathrm{E}}}}(s_1; \mathbf{c}_{\text{true}}) = \sum_{h=1}^{H}\left\langle \mathbf{\Phi}^{\mathsf{T}} d_h^{\pi_{\mathrm{E}}} - \mathbf{\Phi}^{\mathsf{T}} d_h^{\widehat{\pi_{\mathrm{E}}}}, w_{\text{true},h}\right\rangle
$$

$$
\leq H\sqrt{d}\max_{h\in[H]}\left\|\mathbf{\Phi}^{\mathsf{T}} d_h^{\pi_{\mathrm{E}}} - \mathbf{\Phi}^{\mathsf{T}} d_h^{\widehat{\pi_{\mathrm{E}}}}\right\|_{\infty}
$$

$$
\leq H\sqrt{\frac{2d\log(2d/\delta)}{\tau_E}}
$$

where the last inequality holds with probability $1 - \delta$. Therefore, the choice of $\tau_E$ in the theorem statement ensures that $V_1^{\pi_{\mathrm{E}}}(s_1; \mathbf{c}_{\text{true}}) - V_1^{\widehat{\pi_{\mathrm{E}}}}(s_1; \mathbf{c}_{\text{true}}) \leq \epsilon_E$. $\qquad\square$

**Lemma 8.** *For $\beta = \mathcal{O}\left(dH\log(\frac{dT}{\delta})\right)$, the estimator used in Algorithm 4*

$$
\mathbf{v}_h^k = \left(\Lambda_h^k\right)^{-1}\sum_{l=1}^{k}\phi(s_h^l, a_h^l)V_h^k(s_{h+1}^l)
$$

*satisfies for any $h, k \in [H] \times [K]$ and for any state action pair $(s, a) \in \mathcal{S} \times \mathcal{A}$.*

$$
\left|\phi(s, a)^{\mathsf{T}}\mathbf{v}_h^k - P_h V_{h+1}^k(s, a)\right| \leq \beta\left\|\phi(s, a)\right\|_{(\Lambda_h^k)^{-1}} \tag{8}
$$

*with probability $1 - \delta$.*

*Proof.* With analogous steps to the proof of Lemma 5 that

$$
\left|\phi(s, a)^{\mathsf{T}}\mathbf{v}_h^k - P_h V_{h+1}^k(s, a)\right| \leq \left\|M V_{h+1}^k\right\|_{(\Lambda_h^k)^{-1}}\left\|\phi(s, a)\right\|_{(\Lambda_h^k)^{-1}}
$$

$$
+ \left\|\sum_{l=1}^{k}\phi(s_h^l, a_h^l)\left(V_{h+1}^k(s_{h+1}^l) - P_h V_{h+1}^k(s_h^l, a_h^l)\right)\right\|_{(\Lambda_h^k)^{-1}}\left\|\phi(s, a)\right\|_{(\Lambda_h^k)^{-1}}
$$

Then, using the fact that by assumption on $M$ and by the clipping of the value function, we have that $\left\| MV_{h+1}^k \right\|_{(\Lambda_h^k)^{-1}} \leq H$. Then, using (Jin et al., 2019, Lemma B.3) it holds that with probability $1 - \delta$

$$\left\| \sum_{l=1}^k \phi(s_h^l, a_h^l) \left( V_{h+1}^k(s_{h+1}^l) - P_h V_{h+1}^k(s_h^l, a_h^l) \right) \right\|_{(\Lambda_h^k)^{-1}} \leq \mathcal{O}\left( dH \log\left(\frac{dK}{\delta}\right) \right)$$

$\square$

Then, noticing that this is the main term we conclude that

$$\left| \phi(s,a)^\intercal \mathbf{v}_h^k - P_h V_{h+1}^k(s,a) \right| \leq \mathcal{O}\left( dH \log\left(\frac{dK}{\delta}\right) \right) \|\phi(s,a)\|_{(\Lambda_h^k)^{-1}}.$$

**Lemma 9.** *It holds that with probability* $1 - \delta$

$$\sum_{k=1}^K \sum_{h=1}^H \mathbb{E}_{s,a \sim d_h^{\pi^k}} \left[ b_h^{k-1}(s,a) \right] \leq \mathcal{O}\left( d^{3/2} H^2 \sqrt{K \log\left(\frac{2K}{\delta}\right)} \right)$$

*Proof.* We have that

$$2 \sum_{k=1}^K \sum_{h=1}^H \mathbb{E}_{s,a \sim d_h^{\pi^k}} \left[ b_h^{k-1}(s,a) \right] \leq 2 \sum_{k=1}^K \sum_{h=1}^H b_h^{k-1}(s_h^k, a_h^k) + \beta H \sqrt{K \log \delta^{-1}} \quad \text{(Azuma-Hoeffding)}$$

$$\leq 2 \sum_{h=1}^H \sqrt{K \sum_{k=1}^K (b_h^{k-1}(s_h^k, a_h^k))^2} + \beta H \sqrt{K \log \delta^{-1}} \quad \text{(Cauchy-Schwartz)}$$

$$= 2\beta \sum_{h=1}^H \sqrt{K \sum_{k=1}^K \phi(s_h^k, a_h^k)^\intercal (\Lambda_h^{k-1}) \phi(s_h^k, a_h^k)} + \beta H \sqrt{K \log \delta^{-1}}$$

$$\leq 2\beta \sum_{h=1}^H \sqrt{dK \log(2K)} + \beta H \sqrt{K \log \delta^{-1}}$$

$$= \mathcal{O}\left( d^{3/2} H^2 \sqrt{K \log\left(\frac{2K}{\delta}\right)} \right)$$

$\square$

# G EXPERIMENTS

## G.1 EXPERIMENTS WITH DETERMINISTIC EXPERT

We also run an experiment where the expert is deterministic and see if ILARL can compete with BC in this setting. The results are provided in Figure 2. The parameter $\sigma$ is the probability at which the system does not evolve according to the agent's action but in an adversarial way. We experiment with $\sigma = \{0, 0.05, 0.1\}$. The details about the transition dynamics are given in Appendix G.2. Form Figure 2, we can see that ILARL and REIRL are again the most efficient algorithms in terms of MDP trajectories and they are able to match the performance of behavioural cloning despite the fact it has better guarantees for the case of deterministic experts. For Figure 1, we used $\sigma = 0.1$.

## G.2 ENVIRONMENT DESCRIPTION

We run the experiment in the following MDP with continuous states space. We consider a 2D environment, where we denote the horizontal coordinate as $x \in [-1, 1]$ and vertical one as $y \in [-1, 1]$. The agent starts in the upper left corner, i.e., the coordinate $[-1, 1]^\intercal$ and should learn to reach the opposite corner (i.e. $[1, -1]^\intercal$) while avoiding the central high cost area depicted in Figure 3.

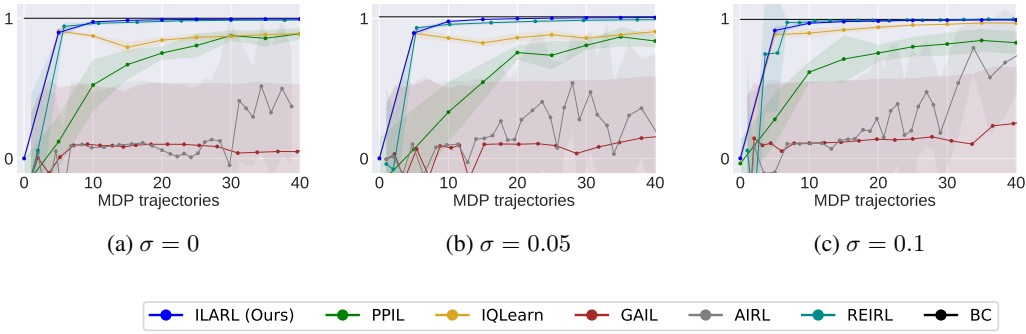

Figure 2: Experiments on the continuous gridworld with one trajectory from a deterministic expert.

The reward function is given by: $\mathbf{c}_{\text{true}}(s,a) = \mathbf{c}_{\text{true}}([x,y]^{\mathsf{T}}, a) = (x-1)^2 + (y+1)^2 + 80 \cdot e^{-8(x^2+y^2)} - 100 \cdot \mathbb{1}\{x \in [0.95, 1], y \in [-1, -0.95]\}$. The action space for the agent is given by

$$\mathcal{A} = \left\{ \underbrace{[0.01, 0]^{\mathsf{T}}}_{\triangleq A_1}, \underbrace{[0, 0.01]^{\mathsf{T}}}_{\triangleq A_2}, \underbrace{[-0.01, 0]^{\mathsf{T}}}_{\triangleq A_3}, \underbrace{[0, -0.01]^{\mathsf{T}}}_{\triangleq A_4} \right\}, \text{ and the transition dynamics are given by:}$$

$$s_{t+1} = \begin{cases} \Pi_{[-1,1]^2}\left[s_t + \frac{a_t}{10}\right] & \text{w.p.} \quad 1-\sigma \\ \Pi_{[-1,1]^2}\left[s_t - \frac{s_t}{10\|s_t\|_2}\right] & \text{w.p.} \quad \sigma \end{cases}$$

Thus, with probability $\sigma$, the environment does not respond to the action taken by the agent, but

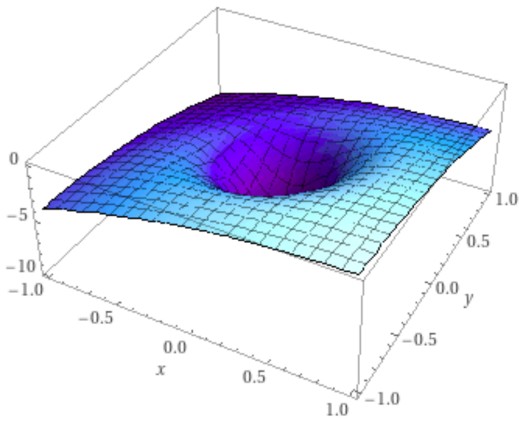

Figure 3: Graphical representation of $-\mathbf{c}_{\text{true}}$ of the linear MDP used in Figures 1 and 2.

it takes a step towards the low reward area centered at the origin, i.e., $-\frac{s_t}{10\|s_t\|_2}$. The agent should therefore pass far enough from the origin. Consider

$$\phi(s,a) = \phi([x,y], a) = \left[x^2, y^2, x, y, e^{-8(x^2+y^2)}, \mathbb{1}\{x \in [0.95, 1], y \in [-1, -0.95]\}, \mathbf{e}_a^{\mathsf{T}}\right]$$

with

$$\mathbf{e}_a = [\mathbb{1}\{a = A_1\}, \mathbb{1}\{a = A_2\}, \mathbb{1}\{a = A_3\}, \mathbb{1}\{a = A_4\}]^{\mathsf{T}}.$$

Notice that Assumption 2 holds only for the cost $\mathbf{c}_{\text{true}} = \mathbf{\Phi}[1, 1, -2, -2, 80, -100, 2, 2, 2, 2]^{\mathsf{T}}$ while for the dynamics the linearity assumption does not hold.

### G.3 NUMERICAL VERIFICATION OF THE FINITE HORIZON IMPROVEMENT.

We test BRIG (Algorithm 4) in a toy finite horizon problem. In particular, we consider a linear bandits problem ($H = 1$) with true cost function $\mathbf{c}_{\text{true}} = \mathbf{\Phi} w_{\text{true}}$ where $\mathbf{\Phi}$ entries are sampled from a normal distribution. For $w_{\text{true}}$ we choose $w_{\text{true}}(i) = 0$ if $i$ is odd and $w_{\text{true}}(i) = 1$ otherwise. We generate the expert dataset sampling 10 actions from a softmax expert. The results are shown in Figure 4. They confirm the theoretical findings that BRIG outperforms ILARL for finite horizon problems in terms of MDP trajectories.

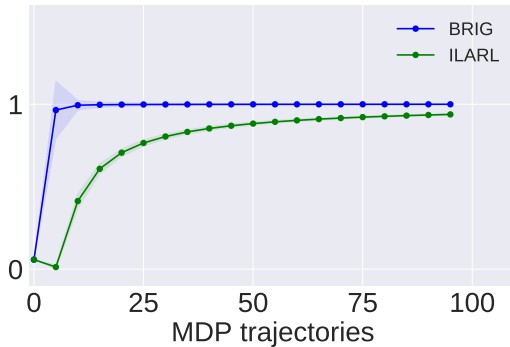

Figure 4: Experiment in finite horizon setting to assess the better efficiency of BRIG.

### G.4 HYPERPARAMETERS

For the experiments in Figures 1 and 2 we used $\eta = 1$, $\tau = 5$ and $\beta = 8$. For IQlearn, we also collect 5 trajectory to perform each update on the $Q$-function, and we use again $\eta = 1$ and $0.005$ as stepsize for the $Q$-function weights. For PPIL, we use batches of 5 trajectories, 20 gradient updates between each batch collection, $\eta = 1$ and and $0.005$ as stepsize for the $Q$-function weights. For GAIL and AIRL, we use the default hyperparameters in `https://github.com/Khrylx/PyTorch-RL` but we obtained a better prerformance with a larger batch size of $6144$ states and we use linear models rather than neural networks. For REIRL, we used the implementation in Viano et al. (2021) but again we increased the batch size equal to $6144$ states for achieving a better performance.

## H REDUCING THE NUMBER OF EXPERT TRAJECTORIES.

In this section, we show that the number of required expert trajectories can be further reduced at the price of additional assumption on the expert policy, features and expert occupancy measure. The estimator we use is build on the ideas underling Mimic-MD in the linear case Rajaraman et al. (2021).

**Remark 4.** *Using such an estimator in ILARL or BRIG allows to improve upon Mimic-MD in two ways. Indeed ILARL and BRIG are provably computationally efficient algorithms and do not require knowledge of the dynamics. On the other hand, Mimic-MD requires perfect knowledge of the transition dynamics and it is unclear if the output policy can be computed efficiently in Linear MDPs.*

To this goal, we need to consider the following estimator for $\mathbf{\Phi}^\intercal d^{\pi_{\text{E}}}$, where we denote via $(s_h^\tau, a_h^\tau)$ the state action pair encountered at step $h$ in the trajectory $\tau$

$$\widetilde{\mathbf{\Phi}^\intercal d^{\pi_{\text{E}}}} = (1 - \gamma)\mathbb{E}_{\tau \sim \pi_{\text{E}}} \left[ \sum_{h=1}^{\infty} \gamma^h \phi(s_h^\tau, a_h^\tau) \mathbb{1}\left[s_t \in \mathcal{K} \quad \forall s_t \in \tau\right] \right]$$

$$+ (1 - \gamma)\mathbb{E}_{\tau \sim \text{Unif}(D_1)} \left[ \sum_{h=1}^{\infty} \gamma^h \phi(s_h^\tau, a_h^\tau) \mathbb{1}\left[\exists s_h \in \tau \ s.t. \ s_h \notin \mathcal{K}\right] \right] \quad (9)$$

where we split the expert dataset $\mathcal{D}_{\pi_{\text{E}}}$ in two disjoint halves $D_0, D_1$. The first $D_0$ is used to compute the set $\mathcal{K}$ which is according to (Rajaraman et al., 2021, Definition 7) the set where the policy

$\pi_{BC}$ learned via Behavioural Cloning on the input dataset $D_0$ equals the expert policy. That is, $\mathcal{K} = \{s \in \mathcal{S} \;\; s.t. \;\; \pi_{BC}(s) = \pi_{\mathrm{E}}(s)\}^3$. The other half denoted via $D_1$ is used for the second term in 9. In the analysis of Rajaraman et al. (2021) the first term can be computed thanks to the perfect knowledge of the dynamics. In our case, we have only trajectory access so we use the estimator

$$\underline{\boldsymbol{\Phi}^{\mathsf{T}} d^{\pi_{\mathrm{E}}}} = (1 - \gamma)\mathbb{E}_{\tau \sim \mathrm{Unif}(\mathcal{D}_{\pi_{BC}})}\left[\sum_{h=1}^{\infty} \gamma^h \phi(s_h^\tau, a_h^\tau)\mathbb{1}\left[s_t \in \mathcal{K} \quad \forall s_t \in \tau\right]\right]$$
$$+ (1 - \gamma)\mathbb{E}_{\tau \sim \mathrm{Unif}(D_1)}\left[\sum_{h=1}^{\infty} \gamma^h \phi(s_h^\tau, a_h^\tau)\mathbb{1}\left[\exists s_h \in \tau \quad s.t. \quad s_h \notin \mathcal{K}\right]\right] \quad (10)$$

where the dataset $\mathcal{D}_{\pi_{BC}}$ contains trajectories sampled according to $\pi_{BC}$.

**Lemma 10.** *Let us consider the estimator $\underline{\boldsymbol{\Phi}^{\mathsf{T}} d^{\pi_{\mathrm{E}}}}$ with the set $\mathcal{K}$ be the confidence set for a binary linear classifier as defined in (Rajaraman et al., 2021, Section 4.1), let the expert policy be deterministic ans satisfy the Linear Expert Assumption (Rajaraman et al., 2021, Definition 4 ).Moreover consider features that satisfy $-\phi(s, 1) = \phi(s, 0) = s/2$ for all $s \in \mathcal{S}$ where the state space is chosen to be $\mathbb{R}^d$. Finally, let consider that $\sum_{a \in \mathcal{A}} d^{\pi_{\mathrm{E}}}(\cdot, a)$ is the uniform distribution $\mathrm{Unif}(\mathcal{S})$, then it holds that for any $\delta > 0$*

$$\mathbb{E}\left\|\underline{\boldsymbol{\Phi}^{\mathsf{T}} d^{\pi_{\mathrm{E}}}} - \boldsymbol{\Phi}^{\mathsf{T}} d^{\pi_{\mathrm{E}}}\right\|_\infty \leq \frac{1}{1 - \gamma}\sqrt{\frac{\log(d/\delta)}{2\left|\mathcal{D}_{\pi_{BC}}\right|}} + \frac{\delta}{1 - \gamma} + \mathcal{O}\left(\frac{d^{5/4}\log d}{(1 - \gamma)^{3/2}\left|\mathcal{D}_{\pi_{\mathrm{E}}}\right|}\right) \quad (11)$$

**Remark 5.** *The Lemma above follows the construction in Rajaraman et al. (2021) to show that there exists one example under which ILARL used with estimator $\underline{\boldsymbol{\Phi}^{\mathsf{T}} d^{\pi_{\mathrm{E}}}}$ requires only $\widetilde{\mathcal{O}}(d^{5/4}\epsilon^{-1}(1 - \gamma)^{-3/2})$ expert trajectories. However, it remains open to prove that the same holds true for general expert in Linear MDPs without further assumptions on the features and expert occupancy measure.*

*Proof.* The error can be controlled as follow

$$E \triangleq \mathbb{E}_{\tau \sim \pi_{\mathrm{E}}}\left[\sum_{h=1}^{\infty} \gamma^h \phi(s_h^\tau, a_h^\tau)\mathbb{1}\left[s_t \in \mathcal{K} \quad \forall s_t \in \tau\right]\right] - \mathbb{E}_{\tau \sim \mathrm{Unif}(\mathcal{D}_{\pi_{BC}})}\left[\sum_{h=1}^{\infty} \gamma^h \phi(s_h^\tau, a_h^\tau)\mathbb{1}\left[s_t \in \mathcal{K} \quad \forall s_t \in \tau\right]\right]$$

so denoting $X(\tau) \triangleq \sum_{h=1}^{\infty} \gamma^h \phi(s_h^\tau, a_h^\tau)\mathbb{1}\left[s_t \in \mathcal{K} \quad \forall s_t \in \tau\right]$ and noticing that by definition of $\mathcal{K}$ we have that

$$\mathbb{E}_{\tau \sim \mathrm{Unif}(\mathcal{D}_{\pi_{BC}})}\left[\sum_{h=1}^{\infty} \gamma^h \phi(s_h^\tau, a_h^\tau)\mathbb{1}\left[s_t \in \mathcal{K} \quad \forall s_t \in \tau\right]\right] = \mathbb{E}_{\tau \sim \mathrm{Unif}(\mathcal{D}_{\pi_{\mathrm{E}}})}\left[\sum_{h=1}^{\infty} \gamma^h \phi(s_h^\tau, a_h^\tau)\mathbb{1}\left[s_t \in \mathcal{K} \quad \forall s_t \in \tau\right]\right],$$

we can rewrite $E$ as a martingale difference sequence

$$\|E\|_\infty = \left\|\mathbb{E}_{\tau \sim \pi_{\mathrm{E}}}\left[X(\tau)\right] - \frac{1}{\left|\mathcal{D}_{\pi_{BC}}\right|}\sum_{\tau \in \mathcal{D}_{\pi_{BC}}} X(\tau)\right\|_\infty$$

$$= \left\|\frac{1}{\left|\mathcal{D}_{\pi_{BC}}\right|}\sum_{\tau \in \mathcal{D}_{\pi_{BC}}}\left(X(\tau) - \mathbb{E}_{\tau \sim \pi_{\mathrm{E}}}\left[X(\tau)\right]\right)\right\|_\infty$$

$$\leq \frac{1}{1 - \gamma}\sqrt{\frac{\log(d/\delta)}{2\left|\mathcal{D}_{\pi_{BC}}\right|}} \quad \text{w.p.} \quad 1 - \delta$$

Therefore choosing $\left|\mathcal{D}_{\pi_{BC}}\right| = \frac{\log(d/\delta)}{2\epsilon^2(1 - \gamma)^2}$ ensures $E \leq \epsilon$ with probability at least $1 - \delta$. Therefore by Lemma 6,

$$\mathbb{E}\|E\|_\infty \leq \frac{1}{1 - \gamma}\sqrt{\frac{\log(d/\delta)}{2\left|\mathcal{D}_{\pi_{BC}}\right|}} + \frac{\delta}{1 - \gamma}$$

---

[3]Notice that we consider a deterministic expert in this section as done in Rajaraman et al. (2021). Therefore, we consider policies as mapping from states to actions, i.e. $\pi : \mathcal{S} \to \mathcal{A}$

These trajectories can be simulated in the MDP therefore the latter it is not a requirement on the expert dataset size. The number of expert trajectories is crucial to control the error due to the trajectories containing trajectories not in $\mathcal{K}$, i.e. Equation (10). Denoting this error as $E_2$ we have

$$\mathbb{E}_{D_0, D_1} \|E_2\|_\infty$$

$$= \mathbb{E}_{D_0, D_1} \left[ \left\| \mathbb{E}_{\tau \sim \pi_E} \left[ \sum_{h=1}^\infty \gamma^h \phi(s_h^\tau, a_h^\tau) \mathbb{1}\left[\exists s_h \in \tau \ s.t. \ s_h \notin \mathcal{K}\right] \right] \right. \right.$$

$$\left. \left. - \mathbb{E}_{\tau \sim \mathrm{Unif}(D_1)} \left[ \sum_{h=1}^\infty \gamma^h \phi(s_h^\tau, a_h^\tau) \mathbb{1}\left[\exists s_h \in \tau \ s.t. \ s_h \notin \mathcal{K}\right] \right] \right\|_\infty \right]$$

$$\leq \frac{1}{(1-\gamma)} \sqrt{\frac{d}{|D_1|} \mathbb{E}_{\tau \sim D_0} \left[ \mathbb{1}\left[\exists s_h \in \tau \ s.t. \ s_h \notin \mathcal{K}\right] \right]}$$

$$= \frac{1}{(1-\gamma)} \sqrt{\frac{d}{|D_1|} \mathbb{E}_{\mathrm{lenght}(\tau)} \left[ \mathbb{E}_{\tau \sim D_0 | \mathrm{lenght}(\tau)} \left[ \mathbb{1}\left[\exists s_h \in \tau \ s.t. \ s_h \notin \mathcal{K}\right] \right] \right]} \quad \text{Tower Property of Expectation}$$

$$\leq \frac{1}{(1-\gamma)} \sqrt{\frac{d}{|D_1|} \mathbb{E}_{\mathrm{lenght}(\tau)} \left[ \sum_{h=1}^{\lceil \mathrm{lenght}(\tau) \rceil} \mathbb{E}_{\tau \sim D_0 | \mathrm{lenght}(\tau)} \left[ \mathbb{1}\left[s_h \notin \mathcal{K}\right] \right] \right]} \quad \text{Union Bound}$$

$$\leq \frac{1}{(1-\gamma)} \sqrt{\frac{d}{|D_1|} \mathbb{E}_{\mathrm{lenght}(\tau)} \left[ \sum_{h=1}^{\lceil \mathrm{lenght}(\tau) \rceil} \mathcal{O}\left( \frac{d^{3/2} \log d}{|D_0|} \right) \right]} \quad \text{Thanks to (Rajaraman et al., 2021, Theorem 7)}$$

$$\leq \mathcal{O}\left( \frac{1}{(1-\gamma)} \sqrt{\frac{d^{5/2} \log d}{|D_1|^2} \mathbb{E}_{\mathrm{lenght}(\tau)} \left[ \mathrm{lenght}(\tau) \right]} \right) \quad \text{Using that } |D_0| = |D_1| \text{ by construction}$$

$$\leq \mathcal{O}\left( \frac{1}{(1-\gamma)} \sqrt{\frac{d^{5/2} \log d}{|D_1|^2} \frac{1}{1-\gamma}} \right)$$

$$= \mathcal{O}\left( \frac{d^{5/4} \log d}{(1-\gamma)^{3/2} |D_1|} \right)$$

Where we used (Rajaraman et al., 2021, Theorem 7) to bound

$$\mathbb{E}_{\tau \sim D_0 | \mathrm{lenght}(\tau)} \left[ \mathbb{1}\left[s_h \notin \mathcal{K}\right] \right] \leq \mathcal{O}\left( \frac{d^{3/2} \log d}{|D_0|} \right)$$

so overall

$$\mathbb{E}_{D_0, D_1} \|E_2\|_\infty \leq \mathcal{O}\left( \frac{d^{5/4} \log d}{(1-\gamma)^{3/2} |D_1|} \right)$$

$\square$

