# OpenReview forum: "Better Imitation Learning in Discounted Linear MDP"
_ICLR.cc/2024/Conference — Submitted to ICLR 2024_

### Official Review · Reviewer_B5xB · 2023-10-27

**Soundness:** 3 good
**Presentation:** 2 fair
**Contribution:** 2 fair
**Rating:** 5
**Confidence:** 4

**Summary:**

This paper aims to design better provably imitation learning (IL) algorithms in discounted linear MDP. In discounted linear MDP, the existing provably IL method PPIL ignores the exploration issue and thus requires the persistent excitation assumption. This paper presents a new method ILARL which is free of such an assumption. In particular, the key to removing such an assumption is the reduction of IL to online optimization with adversarial losses. With this reduction, the target becomes to design a provably RL algorithm in adversarial MDPs. To achieve this goal, this paper first presents an algorithm in the finite-horizon adversarial MDPs and then extends it to the infinite-horizon case. Finally, this paper plugs this RL algorithm into the IL framework, which yields the ILARL algorithm. The authors prove that ILARL has better theoretical guarantees than previous algorithms regarding the number of expert trajectories and MDP trajectories.

**Strengths:**

1. This paper presents a new IL algorithm ILARL and conducts a rigorous theoretical analysis. Compared with the previous SOTA IL method in discounted linear MDP, ILARL removes the persistent excitation assumption and attains better theoretical guarantees on the number of expert trajectories and MDP trajectories.
2. The paper is well-written and easy to follow, providing clear explanations and detailed descriptions of the proposed method and theoretical analysis.

**Weaknesses:**

1. The algorithmic designs and analysis techniques in this paper are not new. In terms of algorithmic designs, the main difference between ILARL, and existing IL algorithms OAL and OGAIL is the policy optimization step. However, the policy optimization algorithm in ILARL is highly similar to the one in [Sherman et al., 2023b].
For theoretical analysis, the key step to removing the persistent excitation assumption is the regret decomposition in Eq.(2), which reduces IL to online optimization with adversarial losses. However, such a regret decomposition has been presented in OAL. Furthermore, among the three types of errors, policy regret is the most difficult part to analyze. However, the analysis of the policy regret in Theorem 3 largely depends on existing techniques developed in OAL and [Sherman et al., 2023b].
2. The empirical evaluation is limited. This paper only considers a simple 2D environment. It is expected to verify the effectiveness of ILARL on more complicated tasks. Besides, this paper does not involve OGAIL for comparison.

**Questions:**

1. Line 9 in Algorithm 3 is a little confusing. Algorithm 2 is a complete RL method that runs for K iterations while line 9 only corresponds to a one-iterate policy update.
2. Typos:
    1. Line 4 in the first paragraph in Section 1: which compete → which competes.
    2. Table 1: OLA → OAL.
    3. Line 14 in Algorithm 1, Line 11 in Algorithm 2: as the cost function is considered, we should minus the bonus function in updating Q functions.

---

> ### Author Response · Authors · 2023-11-15
>
> **The algorithmic designs and analysis techniques in this paper are not new. In terms of algorithmic designs, the main difference between ILARL, and existing IL algorithms OAL and OGAIL is the policy optimization step. However, the policy optimization algorithm in ILARL is highly similar to the one in [Sherman et al., 2023b]. For theoretical analysis, the key step to removing the persistent excitation assumption is the regret decomposition in Eq.(2), which reduces IL to online optimization with adversarial losses. However, such a regret decomposition has been presented in OAL. Furthermore, among the three types of errors, policy regret is the most difficult part to analyze. However, the analysis of the policy regret in Theorem 3 largely depends on existing techniques developed in OAL and [Sherman et al., 2023b].**
>
> We agree that the techniques are similar to the existing works that you mentioned. In particular [Shani et al.,2021] and [Sherman et al., 2023b].  However, notice the following important differences.
>
> 1) The algorithms we present achieves a better regret bound than [Sherman et al., 2023b].We indeed exploit the fact that we have full information about the cost vector while [Sherman et al., 2023b] focuses on the bandit feedback. Moreover our regret bound also improves the result of [Zhong \& Zhang, 2023] for finite horizon adversarial MDPs.
>
> 2) We notice that the regularization in the policy improvement step allows to control the pseudo regret $ \sum^K_{k=1} < d^{\pi_k} - d^{\pi_E}, c_k >$ in the infinite horizon discounted setting.
>
> 3) In the finite horizon case, we obtain a further improvement from $\mathcal{O}(\epsilon^{-4})$ to $\mathcal{O}(\epsilon^{-2})$. This is possible only with the new insight that if we allow the cost player to update first, then, the $\pi$ player can know in advance the cost vector at the next round. We then, notice that LSVI-UCB originally designed for a fixed cost function still guarantees $\mathcal{O}(\sqrt{K})$ regret against a time changing cost vector sequence if the cost vector at the next round can be observed in advance.
> This insight is missing in [Shani et al., 2021] that indeed updates the policy first and then the cost vector. This difference would make LSVI-UCB inapplicable. To fix the situation, the $\pi$ learner could use our Algorithm 1 but this would lead to a MDP trajectories bound of order $\mathcal{O}(\epsilon^{-4})$ which is clearly worst than the $\mathcal{O}(\epsilon^{-2})$ bound attained by our BRIG (Algorithm 4).
>
> In addition, we highlight that the contribution of our submission could be very valuable for the member of the imitation learning community that might not know the power of the connection between imitation learning and online learning in adversarial MDPs.
>
> **The empirical evaluation is limited. This paper only considers a simple 2D environment. It is expected to verify the effectiveness of ILARL on more complicated tasks. Besides, this paper does not involve OGAIL for comparison.**
>
> OGAIL is not implementable because it requires an integration oracle over the state space. See their definition of the features $\phi^h_k$ just after their equation 16 in this version https://proceedings.mlr.press/v162/liu22u/liu22u.pdf
>
> Moreover, OGAIL is not designed for the discounted setting but rather for the finite horizon case.
>
> *Regarding more complicated experiment*
> Given that [Shani et al. 2021, Liu et al. 2022] and our current submission suggests the importance of exploration in imitation learning. Thus,  we are planning to apply exploration heuristics to the policy update in commonly used algorithms like OAL, GAIL and IQLearn and see if the performance is enhanced.
> We think that the answer to this question is currently open. Indeed, in our understanding, the OAL authors provide several MuJoCo experiments but they do not try to implement approximately the optimistic bonuses that are needed to prove the theoretical guarantees.
>
> At the same time, we think that this deep imitation learning extension is beyond the scope of the current submission.
>
> **Line 9 in Algorithm 3 is a little confusing. Algorithm 2 is a complete RL method that runs for K iterations while line 9 only corresponds to a one-iterate policy update.**
>
> Thanks for catching this issue ! We clarified that the policy is updated performing only one iteration of Algorithm 2. We also provide a complete presentation in Algorithm 5.
>
> Thank you also for catching the typos that we corrected in our revision.

---

> > ### Author Response · Authors · 2023-11-21
> >
> > Dear reviewer,
> >
> > Since this is the last day of the discussion phase we would like to ask if you agree with the differences in the techniques with previous work that we pointed out in our answer.
> >
> > In addition, we highlighted that the comparison with OGAIL is not possible because OGAIL is not implementable in our continuous state experiment and we would like to hear your opinion on this point.
> >
> > We hope that our answers improve your feeling about our manuscript and we remain available for further discussion.
> >
> > Best,
> > Authors

---

> > > ### Comment · Reviewer_B5xB · 2023-11-22
> > >
> > > Thanks for your detailed response! However, I think my major concerns about the novelty of algorithmic designs and theoretical analysis still remain.
> > >
> > > **"The algorithms we present achieves a better regret bound than [Sherman et al., 2023b].We indeed exploit the fact that we have full information about the cost vector while [Sherman et al., 2023b] focuses on the bandit feedback. Moreover our regret bound also improves the result of [Zhong & Zhang, 2023] for finite horizon adversarial MDPs."**
> > >
> > > The idea that policy learning in IL can be regarded as solving adversarial MDPs with full information has been proposed in the OAL paper.
> > >
> > > **"We notice that the regularization in the policy improvement step allows to control the pseudo regret in the infinite horizon discounted setting."**
> > >
> > > It seems that the role of policy regularization has been observed in [Moulin & Neu (2023)].
> > >
> > > **"In the finite horizon case, we obtain a further improvement from $\mathcal{O} (\epsilon^{-4})$ to $\mathcal{O} (\epsilon^{-2})$.”**
> > >
> > > The OGAIL paper has already achieved the regret of $\mathcal{O} (\sqrt{K})$ (or sample complexity of $\mathcal{O} (\epsilon^{-2})$) in the finite horizon case. Thus, I think this improvement is not very significant.
> > >
> > > **“In addition, we highlight that the contribution of our submission could be very valuable for the member of the imitation learning community that might not know the power of the connection between imitation learning and online learning in adversarial MDPs.”**
> > >
> > > The connection between IL and online learning in adversarial MDPs has been drawn in the seminal work [Syed & Schapire (2007)] and recent works [Shani et al. (2021)] and [Zahavy et al. (2020)].
> > >
> > > Reference:
> > >
> > > [1] Zahavy et al., "Apprenticeship Learning via Frank-Wolfe." AAAI 2020.

---

> ### Author Response · Authors · 2023-11-22
> **Some important remarks.**
>
> Thanks for your response! We respectfully disagree on some of your remarks.
>
> ***The OGAIL paper has already achieved the regret of $\mathcal{O}(\sqrt{K})$ (or sample complexity of $\mathcal{O}(\epsilon^{-2})$ ) in the finite horizon case. Thus, I think this improvement is not very significant.***
>
> We are worried there is an important misunderstanding. OGAIL achieves the claimed sample complexity for **finite horizon Linear Mixture MDPs**. We do for finite horizon **Linear MDP**.
>
> Despite a similar name, the **Linear MDP** setting is way more involved as the unknown transition dynamics have $d\times |\mathcal{S}|$ degrees of freedom. In **Linear Mixture MDP** those are only $d$.
>
> ***The connection between IL and online learning in adversarial MDPs has been drawn in the seminal work [Syed & Schapire (2007)] and recent works [Shani et al. (2021)] and [Zahavy et al. (2020)].***
>
> We think that the only work that pointed out the same connection is [Shani et al. (2021)]. While the other works are not directly connected.
>
> Indeed, the work of [Zahavy et al. (2020)] requires to compute the optimal policy for each cost vector (see line 5 of their algorithm 1) and it is not discussed how to do it without dynamics knowledge. Similar observation holds for line 7 of Algorithm 1 in [Syed and Shapire, 2007] http://rob.schapire.net/papers/SyedSchapireNIPS2007.pdf
>
> The complication of computing the optimal policy for each intermediate cost function is avoided taking the adversarial MDP perspective of our work and Shani et al. 2021.
>
> ***The idea that policy learning in IL can be regarded as solving adversarial MDPs with full information has been proposed in the OAL paper.***
>
> This is true, but this remark does not seem connected to our claim that you reported in bold before this remark.
> Notice indeed that the work of Shani et al. 2021 holds only for the tabular finite horizon setting while we extend it substantially to the infinite horizon discounted setting in Linear MDP.
>
> So it is unclear how the fact solving adversarial ***tabular*** MDPs with full information has been proposed in the OAL paper makes our contributions in adversarial full information ***Linear*** MDPs less important. Could you please elaborate on this point ?
>
> ***It seems that the role of policy regularization has been observed in [Moulin & Neu (2023)].***
>
> True, we took inspiration from their regularized value iteration algorithm. In our case, we adopted a regularized policy iteration scheme and we noticed that the same observation holds.
> Finally, let us remark that in the context of Linear MDP using regularized policy improvement creates major complications in bounding the covering number of the policy class whereas these are avoided in the Linear Mixture MDP setting of [Moulin & Neu (2023)].
>
>
> Thanks in advance for you attention.
>
>
> Best,
> Authors

---

> > ### Comment · Reviewer_B5xB · 2023-11-23
> >
> > Thank the authors for the detailed response!
> >
> > I'd like to rectify an error in my previous statement. The connection between IL and online learning in adversarial MDPs is drawn in the OAL paper [Shani et al., 2021], instead of the works [Syed & Schapire, 2007, Zahavy et al., 2020].
> >
> > In my understanding, the OAL paper [Shani et al., 2021] first proposes the connection between IL and online learning in adversarial MDPs, and achieves the $\sqrt{K}$ regret in the finite-horizon tabular MDP. The OGAIL [Liu et al., 2021] paper extends OAL to the linear mixture MDP. This submission extends OAL to the linear MDP.
> >
> > Therefore, I find the novelty claim regarding the utilization of full information and the IL-adversarial MDP connection in the initial response to be somewhat less significant because this connection has been drawn in [Shani et al., 2021] and is largely independent of the specific MDP setting. Besides, my concern about evaluating the performance of ILARL on more complicated tasks still remains. Thus I would like to keep my current score unchanged.

---

> > > ### Author Response · Authors · 2023-11-23
> > >
> > > Dear reviewer,
> > >
> > > Thanks for engaging in the discussion and acknowledging the error in your previous response.
> > >
> > > We hope you can also acknowledge that saying that the novelty of our algorithm is limited because it is an extension of OAL to the linear MDP setting is shadowing all the technical difficulties related to controlling the regret in Linear MDP with adversarial losses.
> > >
> > > This problem is very difficult. Way harder than the tabular counterpart that needs to be solved in OAL.
> > >
> > > As matter of fact, it was apparently solved in https://arxiv.org/abs/1912.05830v1 but then the authors realized they had a technical issue that makes the result valid only for linear mixture MDP. Only very recently, there have been correct solutions to this problem for linear MDP ([Zhong & Zhang, 2023, Sherman et al., 2023a] and our submission which also applies to the infinite horizon setting).
> > >
> > > Given that we hope that the reviewer can appreciate the technical difficulties of our work. In particular, we feel that the reviewer is believing that we claim as main contribution the utilization of full information and the IL-adversarial MDP connection (notice that we never claimed this in our initial response) but the main technical difficulty lies in bounding the regret in adversarial linear MDP as explained above.
> > >
> > > Finally, we think that implementing optimistic exploration bonuses beyond linear MDP with continuous states can not be done. We could follow the OAL approach which is implementing the algorithm for Deep Imitation Learning experiments without using the exploration bonuses. However, we feel that it is outside the scope of the current submission.
> > >
> > > Best,
> > > Authors

---

### Official Review · Reviewer_Wgzz · 2023-10-30

**Soundness:** 4 excellent
**Presentation:** 4 excellent
**Contribution:** 3 good
**Rating:** 6
**Confidence:** 3

**Summary:**

The paper proposes an algorithm called ILARL for imitation learning in infinite horizon linear MDP. The authors relax the exploration assumptions in previous works and improve the rate from $O(\epsilon^{-5})$ to $O(\epsilon^{-4}$. The results are built upon a connection between imitation learning and online learning with adversarial lossses. Moreover, the paper presents a strengthen result for the finite horizon case and achieve $O(\epsilon^{-2}$. The empirical results also show that ILARL outperforms other methods.

**Strengths:**

- The paper presents a new algorithm that requires less expert trajectories and MDP trajectories to achieve the $\epsilon$ optimal result in both cases of finite-horizon and infinite-horizon. The results is solid and techniques are novel.
- The paper presents the result and the analysis in a nice way such that it is easy to follow.
- The empirical study also supports the theoretical results about the performance of the proposed algorithm.

**Weaknesses:**

- The paper shows that the learned policy achieve the similar performance as the expert policy. I am wondering if there is any guarantee on the recovery of the true cost function.
- The linear MDP assumption is restrictive. The contribution of the paper can be more significant if it can be extended to general MDPs.
- Although the paper claims that it studies linear MDPs, Assumptions 1-3 are considering the finite state-action case.
- The empirical study is performed on a articrafted MDP rather than a real reinforcement learning environment.

**Questions:**

- Is it possible to extend the result to general MDPs rather than simple linear MDPs?

---

> ### Author Response · Authors · 2023-11-15
>
> Dear reviewer,
>
> Thanks a lot for the words of appreciation to our paper ! In the following we answer your questions.
>
> **The paper shows that the learned policy achieve the similar performance as the expert policy. I am wondering if there is any guarantee on the recovery of the true cost function.**
>
> Not really, we think that recovering the true cost function from a single set of expert demonstration is unfortunately not possible without additional assumptions. There are some recent works along these lines [1,2,3] that we could add in the related works section if the reviewer feels that it would be helpful.
>
> [1] **Reward identification in inverse reinforcement learning**, K Kim et al. - International Conference on Machine Learning, 2021
>
> [2] **Identifiability in inverse reinforcement learning** H Cao et al. - NeurIPS, 2021
>
> [3] **Identifiability and generalizability from multiple experts in Inverse Reinforcement Learning** Rolland et al. - NeurIPS, 2022
>
> **The linear MDP assumption is restrictive. The contribution of the paper can be more significant if it can be extended to general MDPs.**
>
> We have added a new section in the revised manuscript at page 14,15 and 16 where we discuss how the sample complexity of order $\mathcal{O}(\epsilon^{-2})$ for the finite horizon setting can be attained for bilinear classes which is a more general MDP class.
>
> However, the resulting algorithm is not guaranteed to be computationally efficient.
> The focus of our paper was to consider a smaller MDP class but provide a computationally efficient algorithm for this setting.
>
> **Although the paper claims that it studies linear MDPs, Assumptions 1-3 are considering the finite state-action case.**
>
> Thanks for this comment. That's true. We are still considering finite states and action spaces but we assume that the number of states is too large to be enumerated. The exact same proof would go through for infinite states but it would obfuscate the presentation slightly so we preferred to focus on the finite state case.
> On the other end, the action set needs to be finite otherwise we would not have computationally efficient policy updates. For example, the update in Step 20 of Algorithm 1 would require to compute an integral over the action space.
>
> **The empirical study is performed on a articrafted MDP rather than a real reinforcement learning environment.**
>
> We think that the take away of this theoretical work is that exploration in the policy optimization phase can be beneficial for imitation learning.
>
> A natural follow up on the applied imitation learning side is to to try some exploration heuristics for neural networks and insert them in the policy update of common deep imitation learning algorithms like GAIL, IQLearn and see if this would empirically enhanance their performance.
>
> We think that this approach has the potential to lead to convincing empirical performance in more challenging MDPs but at the same times it looks to us beyond the scope of the current submission.
>
> **Is it possible to extend the result to general MDPs rather than simple linear MDPs?**
>
> As mentioned previously, please chack page 14,15 and 16 of the update manuscript for an informal discussion of the extension to bilinear classes.

---

> > ### Author Response · Authors · 2023-11-21
> >
> > Dear reviewer,
> >
> > as the discussion period ends soon we were wondering if our discussion on bilevel classes that we added in the revision solves your question about the extensions to more general MDP classes and if that improves your opinion about our submission.
> >
> > We remain available for further discussion.
> >
> > Best,
> > Authors

---

### Official Review · Reviewer_jGsw · 2023-10-30

**Soundness:** 2 fair
**Presentation:** 2 fair
**Contribution:** 3 good
**Rating:** 5
**Confidence:** 4

**Summary:**

The paper provides a new algorithm for imitation learning under linear MDP setting. By introducing the online learning in MDPs with adversarial losses, the author improves the bound of interactions number with the MDP from $\mathcal{O}(\epsilon^{-5})$ to $\mathcal{O}(\epsilon^{-4})$.

Additionally, unlike previous work, these results do not rely on exploratory assumptions, thereby offering broader applicability.

**Strengths:**

The paper presents a new algorithm, ILARL, namely Imitation Learning via Adversarial Reinforcement Learning algorithm. According to the results in the paper, the required trajectories number in the proposed algorithm has better dependence in $\epsilon$ to achieve the same accuracy. The idea from adversarial online learning is adopted to design this algorithm.

**Weaknesses:**

The paper does not keep consistent notations: $\mathcal{A}$ is used for action space in MDP setting and an algorithm in definition 1; Cost function is utilised in MDP setting, but the numerical experiments adopt reward function setting.

In addition, the paper claims that the ILARL algorithm improves the dependence of accuracy $\epsilon$ from $\mathcal{O}(\epsilon^{-5})$ to $\mathcal{O}(\epsilon^{-4})$, but the dependence of dimension $d$ increases from $d^2$ to $d^3$, so one natural question is that how to carefully select these parameters so that the proposed algorithm indeed requires less trajectories than the latest algorithm in Viano's paper in 2022.

The norm inequalities in assumptions 1-2 seem very technical, and it would be better if the authors could provide some insights about them.

**Questions:**

1. The mathematical formulation for state value function in finite time horizon is pretty strange. I suppose the summation should take from 1 to $h$?

2. The infinite horizon trajectories, according to the description in section 2, have random length sampled from the geometric distribution. Why geometric distribution is adopted here? The sampled number is still finite, so the cost in the time horizons greater than the sampled number is set to zero?

3. In algorithm 3, line 6, the proposed algorithm project $w^{k+1}$ to the unit ball. How to ensure that the projected $w$ still constitute an adversarial costs in $[0, 1]$, as assumed in the MDP setting in section 2? Similar question happens to algorithm 4, line 7.

4. It seems that the proposed algorithms have never updated matrix $M$, in assumption 1 and 2. Does this mean that the true transition kernel is not estimated or involved in the algorithms?

5. The matrix $\Phi$ is already known, according to assumption 1 or 2. But assumption 3 claims that the learner has access to $\Phi$. What is the difference between the matrix $\Phi$ in assumption 3 and $\Phi$ in assumption 1 and 2?

6. As stated in remark 1, the results in theorem 1 and 2 hold with high probability. So theorem 1 and 2 actually state that the trajectories numbers are independent of $\delta$?

7. What is the y-axis in figure 1 and figure 2?

---

> ### Author Response · Authors · 2023-11-13
>
> We thank the reviewer for their time reviewing our paper. Reading your summary, we feel like you are missing the following important contribution. Indeed, we do not only improve the $\epsilon$ dependence from $\mathcal{O}(\epsilon^{-5})$ to $\mathcal{O}(\epsilon^{-4})$ but we also remove **the need for the persistent excitation assumption**.
>
> This is probably an even stronger contribution than improving the $\epsilon$ dependence. Think for example to the special case of tabular MDPs. In this case the features are indicators functions, i.e. $$\phi_{s,a}(s',a') = \begin{cases} 1 \quad \text{if} \quad s,a = s',a' \newline 0 \quad \text{othertwise} \end{cases}$$.
>
> Then, we have that the persistent excitation asumption would require
>
> $$\lambda_{\min} (\mathbb{E}_{s,a \sim d^{\pi^k}} \phi(s,a)\phi(s,a)^T ) \geq \beta > 0$$
>
> Observing that in the tabular case $\phi(s,a)\phi(s,a)^T$ is a diagonal matrix which equals zero everywhere but in the $(s,a)^{\mathrm{th}}$ diagonal element, we conclude that $\mathbb{E}_{s,a \sim d^{\pi^k}} [\phi(s,a)\phi(s,a)^T]$ is also a diagonal matrix where the diagonal elements correspond to the entry of $d^{\pi^k}$.
>
> Therefore, the eigenvalues of the matrix $\mathbb{E}_{s,a \sim d^{\pi^k}}[\phi(s,a)\phi(s,a)^T]$
>
> are the entries of $d^{\pi^k}$ and we can conclude that the persistent excitation assumption amounts to ask that $\min_{s,a} d^{\pi^k}(s,a) > 0$. However, this can be easily contradicted by greedy policies for which at each state there exists only one action such that the above condition holds.
>
> **Cost function is utilised in MDP setting, but the numerical experiments adopt reward function setting.**
>
> Thanks, we will soon update a revision which will make clear that in the experiment section's plot we consider the cumulative reward which equals the opposite of the cumulative cost. Sorry if this created confusion. We made this choice because using costs in the regret analysis is common in the adversarial MDP literature while comparing algorithms with the cumulative rewards is common in imitation learning papers.
>
> **the dependence of dimension $d$ increases from $d^2$ to $d^3$, so one natural question is that how to carefully select these parameters so that the proposed algorithm indeed requires less trajectories than the latest algorithm in Viano's paper in 2022.**
>
> In fact, we can prove that $\frac{1}{\beta} \geq d$. Using this result, we obtain that the dimension dependence in the bound in Viano et al. 2022 is in the best case $d^8$. Therefore, our new algorithm improves the dimension dependence as well.
>
> We show now how to prove that $\frac{1}{\beta} \geq d$
>
> $$
> \beta \leq \lambda_{\min}(\mathbb{E}_{s,a \sim d^{\pi^k}} \phi(s,a)\phi(s,a)^T )
> $$
>
> $$
> \leq \frac{1}{d}\mathrm{Trace} (\mathbb{E}_{s,a \sim d^{\pi^k}}\phi(s,a)\phi(s,a)^T)
> $$
>
> $$
> = \frac{1}{d} \mathbb{E}_{s,a \sim d^{\pi^k}}\mathrm{Trace} (\phi(s,a)\phi(s,a)^T)
> $$
>
> $$
> = \frac{1}{d} \mathbb{E}_{s,a \sim d^{\pi^k}}\mathrm{Trace}(\phi(s,a)^T\phi(s,a))
> $$
>
> $$
> = \frac{1}{d} \mathbb{E}_{s,a \sim d^{\pi^k}} \|\phi(s,a)\|_2^2
> $$
>
> $$
> \leq \frac{1}{d} \mathbb{E}_{s,a \sim d^{\pi^k}} \|\phi(s,a)\|_1^2
> $$
>
> $$
> \leq \frac{1}{d} \max_{s,a} \|\phi(s,a)\|_1^2
> $$
>
> $$
> = \frac{1}{d} \|\phi(s,a)\|_{1,\infty}^2 \leq \frac{1}{d}
> $$
>
> where the last step follows from $\|\phi(s,a)\|_{1,\infty} \leq 1$ as assumed in Assumptions 1 and 2.
>
> **The norm inequalities in assumptions 1-2 seem very technical, and it would be better if the authors could provide some insights about them.**
>
> Assuming bounded features is common in the Linear MDP setting see for example the paper by Jin et al., 2019.
>
> The intuition is that since the quantities that we are assuming to lie in the column span of the matrix $\Phi$ are bounded (the cost and the transition probabilities are bounded by one). Therefore, it would not make sense to use potentially unbounded features to represent the cost and the transition dynamics as an inner product.
>
> **The mathematical formulation for state value function in finite time horizon is pretty strange. I suppose the summation should take from 1 to h?**
>
> There is a small typo, the sum should be from $h$ to $H$ as standard. See for example equation 4.2.1 in Puterman's book.
>
> **The infinite horizon trajectories, according to the description in section 2, have random length sampled from the geometric distribution. Why geometric distribution is adopted here?**
>
> The geometric distribution is the standard choice in Discounted Infinite Horizon MDP (see Chapter 6 in Puterman's book).

---

> ### Author Response · Authors · 2023-11-13
> **Second part of the Answer**
>
> **The sampled number is still finite, so the cost in the time horizons greater than the sampled number is set to zero?**
>
> Notice that in defining the value function $V^{\pi}(s; c) \triangleq \mathbb{E} [\sum^{\infty}_{h=1} \gamma^{h-1} c(s_h,a_h) | s_1 = s] $ there is no sampling involved and since the geometric distribution is unbounded we can not set to zero some of the costs.
>
> However, in the algorithms we use finite values as horizon sampled from the geometric distribution and we do not need to observe the costs for stages larger than the sampled value.
> We clarify this in Protocol 1 in Appendix B.
>
> **It seems that the proposed algorithms have never updated matrix $M$, in assumption 1 and 2. Does this mean that the true transition kernel is not estimated or involved in the algorithms?**
>
> Interesting question! Indeed, the matrix $M$ is never estimated by the Algorithm. The problem in doing this is that the matrix $M$ has dimensions $d \times S $. Therefore, estimating it would unavoidably lead to an algorithm which requires memory linear in $S$. To avoid such an issue our algorithm only tries to estimate the quantity $M V^k$ where $V^k$ are the value functions sequence produced by the Algorithm.
>
> The advantage of this approach is that $M V^k$ is just a $d$ dimensional vector so our algorithm enjoys memory and running time which is independent of $S$.
>
> **The matrix  Φ is already known, according to assumption 1 or 2. But assumption 3 claims that the learner has access to Φ. What is the difference between the matrix Φ in assumption 3 and Φ in assumption 1 and 2?**
>
> The difference is in which cost functions we require to be linear in the features. In Assumption 1 and 2 we require the sequence of changing costs to be linear in the features. Assumption 3, which is needed in the imitation learning setting, we additionaly assume that the true unknown cost function is linear in the features.
>
>
> **As stated in remark 1, the results in theorem 1 and 2 hold with high probability. So theorem 1 and 2 actually state that the trajectories numbers are independent of δ ?**
>
> The dependence on $\delta$ does not appear in Theorem 1 and 2 because we are stating the result in expectation. As highlighted in Remark 1, we prove this result passing first via a high probability result (Theorems 5 and 6) and then as shown in Lemma 6 we set $\delta = \epsilon$ to obtain the result given in Theorem 1 and 2.
>
> Check Theorem 5 and 6 to appreciate how the number of required trajectory depend on $\delta$.
>
> **What is the y-axis in figure 1 and figure 2?**
>
> It is the normalized cumulative return. We normalize it in a way that the expert performance averaged over $10$ seeds equals 1 and the one of the uniform policy equals 0.
>
> **In algorithm 3, line 6, the proposed algorithm project $w^{k+1}$ to the unit ball. How to ensure that the projected $w$ still constitute an adversarial costs in [0,1], as assumed in the MDP setting in section 2? Similar question happens to algorithm 4, line 7.**
>
> This is ensured by Assumptions 1 and 2 respectively. In particular, we assume that each adversarial cost $c^k$ can be written as linear combination of the column span of $\Phi$ with weights $w^k$ such that $||w^k||_2 \leq 1$. In the imitation learning contest, projecting in the unit ball ensures this last bound on the weights.
>
>
>
>
> **Thanks again for the interesting questions. We hope that our answer will help you to appreciate our contribution better and improve your assessment of our submission. Furthermore, we remain open to discussion.**
>
> Best,
>
> Authors.

---

> > ### Comment · Reviewer_jGsw · 2023-11-14
> >
> > Thank you for your detailed explanations and pointing out that your contribution in relaxing exploratory assumptions, due to which I improved the rating for contribution and the final rating of the paper.
> >
> > I am still confusing about your answer to the geometric distribution in setting random length, and I did not find it in Chapter 6 in Puterman's book, would you mind providing a more specific place (like, in which section or other materials)?
> >
> > In addition, I am also curious about the projection of $w^{k+1}$ (my third question in the comments), could you please provide the explanations for this?
> >
> >
> > Best,
> > Reviewer

---

> > > ### Author Response · Authors · 2023-11-14
> > >
> > > Dear reviewer,
> > >
> > > Thanks a lot for engaging in the discussion and for your improved evaluation !
> > >
> > > A good discussion on the interpretation of the discounted model setting is actually in section 5.3 in Puterman's book. For your convenience, we attached a screenshot of the relevant paragraph here: https://imgur.com/a/WtiaJAD
> > >
> > > You are right that the projection can not ensure that $c^k \in [0,1]$ but only that $c^k \in [-1,1]$. Indeed we have that since $c^k = \Phi w^k$ it holds that $$c_k \geq -  |\Phi|_{\infty,1} |w^k| _{\infty} \geq -1$$
> > >
> > > and
> > >
> > > $$
> > >  1 \geq |\Phi|_{\infty,1} |w^k| _{\infty} \geq  c^k
> > > $$
> > >
> > > However this discrepancy is easy to fix. In Algorithm 3, we can just shift the resulting cost between $[0,1]$ after the projection as follows $c^k = \frac{\Phi w^k + 1}{2}$ before feeding the cost vector to Algorithm 2. This can be done because affine transformation of the cost vector do not affect the optimality of the policy. Notice indeed that
> > >
> > > $$
> > > \mathrm{argmax}_{\pi} < d^{\pi}, c^k >
> > > $$
> > >
> > > $$
> > > = \mathrm{argmax}_{\pi} < d^{\pi}, \frac{\Phi w^k}{2} >
> > > $$
> > >
> > > $$
> > >  =  \mathrm{argmax}_{\pi} < d^{\pi}, \Phi w^k>
> > > $$
> > >
> > > We hope that these answers clarify your remaining concerns and will further improve your evaluation of the paper.
> > >
> > > Best,
> > > Authors

---

> > > > ### Author Response · Authors · 2023-11-20
> > > >
> > > > Dear reviewer jGsw,
> > > >
> > > > We were wondering if you feel satisfied about the discussion in Puterman's book on the discounted setting and about our answer on the projection step.
> > > >
> > > > Please notice that in our revision, we modified line 8 of algorithm 3 to make sure that Algorithm 2 is invoked with costs bounded between [0,1] and not with costs between [-1,1].
> > > >
> > > > Best,
> > > > Authors

---

> > > > > ### Comment · Reviewer_jGsw · 2023-11-22
> > > > >
> > > > > Dear authors,
> > > > >
> > > > > Thank you for your answer about Puterman’s book and the illustration of projection step.
> > > > >
> > > > > I will keep the current score.

---

> > > > > > ### Author Response · Authors · 2023-11-23
> > > > > >
> > > > > > Dear reviewer,
> > > > > >
> > > > > > Thanks for your response!
> > > > > >
> > > > > > Apparently we clarified all your concerns therefore could you please elaborate on which issues are still preventing you from giving an higher score to our submission ?
> > > > > >
> > > > > >
> > > > > > Best,
> > > > > >
> > > > > > Authors

---

### Official Review · Reviewer_Rc5N · 2023-11-01

**Soundness:** 3 good
**Presentation:** 2 fair
**Contribution:** 3 good
**Rating:** 5
**Confidence:** 4

**Summary:**

The contribution of the paper is to provide a more sample efficient algorithm for both discounted IL in terms of the number of samples of environment interaction and expert trajectories required for the linear MDP setting. The proof involves exploiting a connection between imitation learning and online learning in MDPs with adversarial losses in the full information setting, pointed out by Viano et al (2022). The authors provide experimental evidence on a simple gridworld environment showing the utility of their approach where the algorithm always achieves near-expert policy and often surpasses it.

**Strengths:**

The key idea is to online-to-batch to convert the IL problem to a regret minimization problem, and use a regret decomposition developed in Viano et al (2022) which decomposes the regret into 3 parts: a regret of matching the occupancy measure of the expert under a linear cost c_k, a regret term from approximating the true linear function, $w_{\text{true}}$ capturing the reward function measured against the estimation error of the expert’s occupancy measure, and the last term being the estimation error of the expert’s occupancy measure. The first two terms in the regret decomposition are for learning a policy which performs well on the estimate of the expert’s occupancy measure on the feature space.

An empirical estimate controls the 3rd error term, the second regret term can be controlled by OGD or any other online learning algorithm, while the authors provide an improved analysis for the regret of the first term, which cannot easily be solved by a generic no-regret algorithm because of the unknown transition dynamics, which make it impossible to project onto the space of valid occupancy measures. The authors propose a no regret algorithm in two steps: Policy evaluation is done using a fresh batch of data collected on-policy to get an optimistic estimate of the Q function; policy updates are not done greedily, but are done using the average optimistic $Q$ value computed from a batch of episodes to carry out infrequent policy updates. The approach resembles an MWE update with a finite-buffer to eliminate old and inaccurate $Q$ estimates. In a sense, the approach is similar to variance reduction in stochastic optimization.

The novelty in the analysis in the paper is in showing an improved algorithm for linear MDPs with adversarial costs in the full information setting. The authors show a regret bound which scales as $\tilde{O} (d^{3/4} H^{3/2} K^{3/4})$ which improves over the previous best result of $\tilde{O} (d^{3/4} H^2 K^{3/4})$.

**Weaknesses:**

The paper improves the prior state of the art in the best known sample complexity for IL in the discounted and finite horizon settings, and the technical novelty is moderate. The analysis is largely to improve the best  known results for linear MDPs with adversarial costs in the full information setting. This is novel and may be of independent interest, but largely borrows insights from previous work, Viano et al (2022), the analysis of UCB in Jin et al and analysis of no-regret algorithms (MWE). I think it's still a nice contribution, but feels somewhat like an A+B(+C) type result.

Overall, the writing of the paper is ok, there is room for improvement in terms of the presentation. The related work section can be organized in a much better way. This is important to put into context the results in the paper. There are several lines of work related to this one, and so it's all the more important to structure the related section in a better way.

The experimental eval in the paper is very limited, and what seem to be on a very simple environment. While the theory in the paper is the major contribution, it would have been helpful to see a more comprehensive evaluation. I am not reducing my score for the paper because of this point, but if the paper gets rejected, I encourage the author to run more comprehensive experiments. Typically on harder environments, it is quite difficult to achieve the expert's performance, but this is not the case in any of the experiments in the paper.

The results of Rajaraman et al (2021) in the offline setting do not require linear reward or a uniform occupancy measure. These assumptions seem to be used in the online setting to get improved bounds. In the online setting, the work Swamy et al (2022) provides a general analysis of the estimator used in Lemma 10 of the paper to go beyond the uniform feature measure assumption. While in general these two lines of work are not comparable, since the current paper assumes a model where the expert is arbitrary but the optimal policy falls in a linear class, as opposed to the linear expert setting (where the expert is a linear classifier), it would be interesting to see in a future work if there is a better connection between these settings.

Swamy et al (2022): https://proceedings.neurips.cc/paper_files/paper/2022/file/2e809adc337594e0fee330a64acbb982-Paper-Conference.pdf

Minor:
1. "Therefore, the policy suboptimality scale as $H^4 \log |\Pi| / \epsilon^2$". Isn’t the policy suboptimality precisely $\epsilon$?

**Questions:**

1. The standard BC reduction, for the finite horizon setting argues that BC achieves a $O(H^2)$ suboptimality scaling in finite-horizon settings. This is in contrast to the discussion in the discounted setting on page 1 for BC. I am not aware of a work or analysis which states that BC requires $\widetilde{O} (1/(1-\gamma)^4)$ demonstrations in the discounted setting. It would be helpful to cite a paper here.

2. Do the results in this setting hold beyond linear MDPs, say for bilinear classes or Bellman rank bounded MDPs? It would have been nice to include a discussion about this point.

---

> ### Author Response · Authors · 2023-11-15
>
> Dear reviewer,
>
> thanks a lot for the time spent reviewing our work !
>
> We would like to start the rebuttal with 2 clarification. First, we would like to emphasize the fact that the improvement is not only on the dependence on the dimension or the accuracy $\epsilon$ but it is on avoiding the limiting persistent excitation assumption.
> This is very important as we explain in our common answer.
>
> Second, the connection to adversarial MDP was not pointed out in Viano et al. 2022. It appeared in (Shani et al., 2021) for the finite horizon tabular setting and then it went somehow forgotten in the imitation learning community. In this work, we adapt it to finite and infinite horizon Linear MDP. New insights are required when compared to (Shani et al., 2021) to prove the improved bound in the finite horizon setting.
>
>
> **The paper improves the prior state of the art in the best known sample complexity for IL in the discounted and finite horizon settings, and the technical novelty is moderate. The analysis is largely to improve the best known results for linear MDPs with adversarial costs in the full information setting. This is novel and may be of independent interest, but largely borrows insights from previous work, Viano et al (2022), the analysis of UCB in Jin et al and analysis of no-regret algorithms (MWE). I think it's still a nice contribution, but feels somewhat like an A+B(+C) type result.**
>
> Please also appreciate the fact that there are some key differences when compared with the ideas used in the UCB analysis of Jin et al. . In particular, their analysis crucial relies on the fact that they can prove that for every $k$, and for every state action pair $s,a$ and for every stage $h$, it holds that $Q_h^k(s,a) > Q_h^\star(s,a)$ (upper bound because in Jin's paper they consider rewards instaed of costs) with high probability. In the infinite horizon, unfortunately we can not establish this property so we have to exploit a different form of optimism which is that the $Q$ values are lower bounds to the ideal value iterations updates. That is, $Q^k(s,a) \leq c^k(s,a) + \gamma P V^k(s,a)$ for every state action pair $s,a$ and round $k$ with high probability.
>
> **Overall, the writing of the paper is ok, there is room for improvement in terms of the presentation. The related work section can be organized in a much better way. This is important to put into context the results in the paper. There are several lines of work related to this one, and so it's all the more important to structure the related section in a better way.**
>
> We are happy to work to improve the writing further! Which lines of work do you feel we are missing ?

---

> ### Author Response · Authors · 2023-11-15
>
> **The experimental eval in the paper is very limited, and what seem to be on a very simple environment. While the theory in the paper is the major contribution, it would have been helpful to see a more comprehensive evaluation. I am not reducing my score for the paper because of this point, but if the paper gets rejected, I encourage the author to run more comprehensive experiments. Typically on harder environments, it is quite difficult to achieve the expert's performance, but this is not the case in any of the experiments in the paper.**
>
> Thanks for the suggestion. We think that the take away of this theoretical work is that exploration in the policy optimization phase can be beneficial for imitation learning.
>
> A natural follow up that we are planning is to to try some exploration heuristics for neural networks and insert them in the policy update of common deep imitation learning algorithms like GAIL, IQLearn and see if this would empirically enhanance their performance.
>
> **The results of Rajaraman et al (2021) in the offline setting do not require linear reward or a uniform occupancy measure. These assumptions seem to be used in the online setting to get improved bounds. In the online setting, the work Swamy et al (2022) provides a general analysis of the estimator used in Lemma 10 of the paper to go beyond the uniform feature measure assumption. While in general these two lines of work are not comparable, since the current paper assumes a model where the expert is arbitrary but the optimal policy falls in a linear class, as opposed to the linear expert setting (where the expert is a linear classifier), it would be interesting to see in a future work if there is a better connection between these settings.**
>
> This is another very interesting point ! We added a discussion about this in our revision. Please check page 14 of our revised manuscript.
>
> **The standard BC reduction, for the finite horizon setting argues that BC achieves a $O(H^2)$ suboptimality scaling in finite-horizon settings. This is in contrast to the discussion in the discounted setting on page 1 for BC. I am not aware of a work or analysis which states that BC requires $\mathcal{O}((1 - \gamma)^{-4})$ demonstrations in the discounted setting. It would be helpful to cite a paper here.**
>
> Good question. This is a potentially confusing point. The common bound for BC says that the policy suboptimality error for the policy output denoted as $\pi_B$ is bounded as
>
> $$
> V^{\pi_{B}} - V^{\pi_E} \leq (1 - \gamma)^{-2} \mathbb{E}_{s \sim d^{\pi_E}} \sum_a | \pi_E(a|s) - \pi_B (a|s)|
> $$
>
> so here the scaling is $(1 - \gamma)^{-2}$ as you mentioned. However, for general experts, the dependence become quartic when we ask how many expert trajectories are necessary to achieve $V^{\pi_{B}} - V^{\pi_E} \leq \epsilon$.
> In this case, we can use that by Hoeffding we have that with high probability
> $$
> \mathbb{E}_{s \sim d^{\pi_E}} \sum_a | \pi_E(a|s) - \pi_B(a|s) |  \leq \sqrt{\frac{\log (\Pi/\delta)}{N_E}}
> $$
> where $N_E$ is the number of expert trajectories.
> Therefore
>
> $$
> V^{\pi_{B}} - V^{\pi_E} \leq (1 - \gamma)^{-2} \sqrt{\frac{\log (\Pi/\delta)}{N_E}}
> $$
>
> So, finally to have that the right hand side equal to $\epsilon$ we need $N_E \geq \frac{\log (\Pi/\delta)}{(1 - \gamma)^4\epsilon^2}$.
> In this last bound the horizon dependence shows up with quartic dependence. The same result has been proven in (Li \& Zhang, 2022).
>
>
> **Do the results in this setting hold beyond linear MDPs, say for bilinear classes or Bellman rank bounded MDPs? It would have been nice to include a discussion about this point.**
>
> Yes, we think that we can extended the result to finite horizon bilinear classes ! However, the resulting algorithm can not be ensured to be computationally efficient. We added an informal discussion about this extension at page 14,15 and 16 of our revised draft.
>
> The main focus of our paper is to consider the smaller class of linear MDP but give an efficient algorithm for it.
>
> **Therefore, the policy suboptimality scale as ...**
>
> This is a typo, we meant that the required number of expert demonstrations scale as $\frac{H^4 \log \Pi}{\epsilon^2}$. We corrected this in our revision. Thanks for catching it !

---

> > ### Author Response · Authors · 2023-11-21
> >
> > Dear reviewer,
> >
> > Does the discussion regarding the horizon scaling and the extension to bilinear classes solve your questions ?
> >
> > Hopefully, our answers will improve your assessment of the paper.
> >
> > Best,
> > Authors

---

### Author Response · Authors · 2023-11-15
**On the importance of removing the persistent excitation assumption**

Dear reviewers,

Thank you all for your time spent reading our paper and writing useful comments and interesting suggestions.

We would like to provide a common response to highlight the main contribution of the paper which is to remove the need of the persistent excitation assumption. We are doing this because we noticed that this contribution is sometimes missing from your summary of the paper.

In our opinion, this contribution is even more important than improving the dependence on $\epsilon$.

**Failure of persistent excitation in tabular MDP**

First, notice that the assumption can be easily violated. For example, by a deterministic policy in a tabular MDP.

 In this case the features are indicators functions, i.e. $$\phi_{s,a}(s',a') = \begin{cases} 1 \quad \text{if} \quad s,a = s',a' \newline 0 \quad \text{othertwise} \end{cases}$$.

Then, we have that the persistent excitation asumption would require

$$\lambda_{\min} (\mathbb{E}_{s,a \sim d^{\pi^k}} \phi(s,a)\phi(s,a)^T ) \geq \beta > 0$$

Observing that in the tabular case $\phi(s,a)\phi(s,a)^T$ is a diagonal matrix which equals zero everywhere but in the $(s,a)^{\mathrm{th}}$ diagonal element, we conclude that $\mathbb{E}_{s,a \sim d^{\pi^k}} [\phi(s,a)\phi(s,a)^T]$ is also a diagonal matrix where the diagonal elements correspond to the entry of $d^{\pi^k}$.

Therefore, the eigenvalues of the matrix $\mathbb{E}_{s,a \sim d^{\pi^k}}[\phi(s,a)\phi(s,a)^T]$

are the entries of $d^{\pi^k}$ and we can conclude that the persistent excitation assumption amounts to ask that $\min_{s,a} d^{\pi^k}(s,a) > 0$. However, this can be easily contradicted by greedy policies for which at each state there exists only one action such that the above condition holds.

**Improved dimension dependence in the general case**

Moreover, in the linear case we noticed thanks to the question of Reviewer jGsw that the $\beta$ of the persistent excitation assumption needs to satisfy $\frac{1}{\beta} \geq d $.
For the proof, please check the response to Reviewer jGsw.

This means that in the best case for PPIL, our algorithm still improves drastically the dimension dependence of the bound from $d^8$ to $d^3$.

**Final comment**

Even if in the hindsight the proof techniques might look based on existing results, we think that this is an important contribution for the imitation learning community which is not necessarily aware of the connection between imitation learning and adversarial MDPs and of the exploration techniques used in online learning in adversarial MDPs. We hope you consider the current submission also under this point of view.

We hope that this clarifications and the individual answers to your question will help to improve your evaluation of the paper!

Thanks again for your time!

Best,

Authors

---

### Author Response · Authors · 2023-11-15
**Changes in the revised manuscript**

Dear reviewer,

This message is to help you navigating the revised manuscript that we just uploaded.

First, we corrected all the typos that you noticed. Thanks again for catching them.

Moreover, as suggested by reviewer Rc5N, we have added in page 14 a discussion regarding a comparison and possible future directions inspired by the work [Swamy et al., 2022]

Finally, since Reviewers Rc5N and Wgzz asked about extensions to more general MDPs model for example to bilinear classes, we added in page 14,15,16 how one could control the policy player with the algorithm in [1] to prove a MDP trajectory bound in Bilinear Classes of order $\mathcal{O}(\epsilon^{-2})$.
To obtain this result, we use again the observation that the $\pi$ player can update knowing the next cost vector in advance and we noticed that [1] can be easily modified to handle adversarial costs if they are known one round in advance to the learner.

Unfortunately, the same result does not seem to be provable for the more challenging infinite horizon setting.

Thanks again for your attention,

Best,

Authors.

[1] Bilinear classes: A structural framework for provable generalization in rl, S Du et al., International Conference on Machine Learning, 2021

---

### Author Response · Authors · 2023-11-20

Dear reviewers,

as the end of the discussion phase is approaching soon, we are wondering if our answers solve your doubts regarding the submission.

On our end, we would be happy to provide any additional clarification if needed.

Best,
Authors

---

### Meta-Review · Area_Chair_TDpV · 2023-12-09

**Metareview:**

This paper considers imitation learning in (discounted, infinite-horizon) linear MDPs. It develops algorithms that remove a "persistent excitation" assumption required for MDP trajectory sample complexity bounds in previous work, and provides better expert trajectory and regret bounds for both finite and infinite horizon settings.

The strength of the paper are the improved bounds and technical analysis providing those. Some of the reviewers question the novelty of the approach, viewing this as a combination of existing methods. Additionally, though the paper shows empirical benefits on an continuous gridworld imitation learning task, the reviewers found this to be a limited demonstration of the approach and suggested more challenging settings. The negatives slightly tend to outweigh the positives of the paper, preventing an acceptance recommendation in its current form.

**Justification For Why Not Higher Score:**

The reviewers felt the paper was combining existing techniques, limiting the overall novelty of the contribution, and that the experimental setting was too simplistic to adequately demonstrated the benefits of the approach.

**Justification For Why Not Lower Score:**

N/A

---

### Decision · Program_Chairs · 2024-01-16

Reject